EMBO
Molecular Medicine

# Dynamics of checkpoint receptors in γδ T cell subsets are associated with clinical response during anti-PD-1 immunotherapies

Elisa Catafal-Tardos [1], Lola Dachicourt[1], Maria Virginia Baglioni [1], Marcelo Gregorio Filho Fares da Silva [1], Davide Secci[1], Marco Donia [2], Anders Handrup Kverneland [2], Inge Marie Svane[2] & Vasileios Bekiaris [1]✉

## Abstract

**Gamma delta (γδ) T cells are innate-like lymphocytes with potent anti-tumor properties. Herein, we show that immune checkpoint receptors (ICRs) display differential expression and regulation by the JAK-STAT pathway in Vδ1 and Vδ2 cells and identify constitutive (e.g. TIGIT, PD-1) and inducible (e.g. TIM-3, LAG-3, CTLA-4) ICRs. In melanoma, all γδ T cell subsets downregulated AP-1 transcription factors, but Vδ1 cells specifically expressed high levels of ICR, TOX and inhibitory killer Ig-like receptor (KIR) transcripts, reminiscent of exhaustion. However, patient-derived cells were functionally competent, although induction of LAG-3 and CTLA-4 was impaired. During anti-PD-1 monotherapy, Vδ1 cells specifically bound high levels of therapeutic antibody but only in patients who responded to treatment, revealing a potential new prognostic marker for evaluating the efficacy of IC blockade (ICB) therapy. Finally, expression of KIR genes in Vδ1 cells was downregulated in response to successful ICB therapy. Collectively, our data indicate an intricate relationship between ICRs and γδ T cells and reveal novel approaches by which these cells can be harnessed in order to discern or improve cancer immunotherapy.**

**Keywords** Cancer; Immunotherapy; γδ T Cells; Immune Checkpoint Receptors; Melanoma
**Subject Categories** Cancer; Immunology

## Introduction

Immune checkpoint receptors (ICRs) transmit inhibitory signals to counter-regulate leukocyte activation and reduce the chances of unwanted inflammatory disease. In T cells, most ICRs inhibit pathways downstream of the T cell receptor (TCR) or co-stimulatory molecules, such as CD28 (He and Xu, 2020; Worboys

and Davis, 2024), however, different receptors will engage in diverse mechanisms of signaling. For example, programmed death-1 (PD-1) contains immunoreceptor tyrosine-based inhibitory and immunoreceptor tyrosine-based switch motifs (ITIM and ITSM), which can directly associate with the Src-homology-2 phosphatases 1 and 2 (SHP-1 and SHP-2) to inhibit T cell activity (Patsoukis et al, 2020). Lymphocyte activation gene 3 (LAG-3) is another ICR that utilizes cytoplasmic inhibitory motifs, such as its unique KIEELE motif and a glutamic acid–proline rich tandem repeat (EP) motif, in order to directly suppress T cell responses (Chocarro et al, 2021; Guy et al, 2022). Instead of active inhibitory motifs, the cytoplasmic tail of T cell immunoglobulin and mucin domain-containing protein 3 (TIM-3) interacts with HLA-B associated transcript 3 (Bat3) at steady-state, which sustains T cell activation (Wolf et al, 2020). However, upon ligand binding, Bat3 is released and TIM-3 assumes an inhibitory function leading to T cell death and exhaustion (Rangachari et al, 2012; Wolf et al, 2020). Although cytotoxic T lymphocyte antigen-4 (CTLA-4) utilizes its intracellular inhibitory domain (Rudd et al, 2009), it can bind to CD80 and CD86 with high affinity, and as a result transendocytose them from antigen presenting cells, preventing thus CD28 mediated co-stimulation (Kennedy et al, 2022; Ovcinnikovs et al, 2019; Qureshi et al, 2011). Similar to CTLA-4, T cell immunoreceptor with immunoglobulin and ITIM domain (TIGIT) inhibits both through direct signaling and by disrupting co-stimulation through ligand competition (Tang et al, 2023).

The above inhibitory mechanisms have been successfully explored in cancer therapy, so that antibody(Ab)-mediated blockade of ICRs or their ligands can prevent inhibition and enhance anti-cancer T cell responses, leading in many cases to tumor regression (Postow et al, 2015). Current FDA approved immune checkpoint blockade (ICB) therapies target PD-1, PDL-1, CTLA-4, and LAG-3 (Sun et al, 2023), while a number of clinical trials indicate that soon there will be drugs approved for both TIM-3 and TIGIT (Cai et al, 2023), further enhancing the importance of this cell regulatory system as a targeted therapy. Nevertheless, many patients fail to respond to ICB therapy and of those who initially respond, a significant proportion will develop resistance (Bagchi

[1]Department of Health Technology, Technical University of Denmark, Kgs Lyngby, Denmark. [2]National Center for Cancer Immune Therapy - CCIT-DK, Department of Oncology, Copenhagen University Hospital, Herlev, Denmark. ✉E-mail: vasbek@dtu.dk

et al, 2021; Sharma et al, 2023). Resistance takes various forms including tumor-intrinsic, such as impaired MHC expression, or tumor-extrinsic usually involving the immune system, such as accumulation of immuno-suppressive cells and factors within the tumor microenvironment (TME) (Bagchi et al, 2021; Sharma et al, 2023). In this regard, it is important that we uncover in greater detail the biology of ICRs as well as their mode of action during ICB therapies.

A lot of what we know regarding the benefits of ICB therapy stems from studies in conventional alpha beta (αβ) T cells, who are more prone to resistance mechanisms, including altered or suppressed MHC presentation or the presence or lack of neoantigens (Sharma et al, 2023). However, some patients with alterations in genes associated with class I presentation, such as *B2M*, can still respond to ICB presumably due to the action of unconventional gamma delta (γδ) T cells (de Vries et al, 2023), while cancers with low neoantigen load that are responsive to ICB are characterized by increased intra-tumoral γδ T cell associated transcripts (Davies et al, 2024). These data lead us to hypothesize that γδ T cells may hold a solution towards improving the outcome of ICB therapy (Davies et al, 2024; de Vries et al, 2023). γδ T cells are polyfunctional innate-like lymphocytes conserved in evolution, whose TCR is peptide/MHC-independent and can instead recognize lipids and a variety of self-ligands, such as butyrophilins, annexin A2 or endothelial protein receptor 2 (Marlin et al, 2017; Sebestyen et al, 2019; Willcox and Willcox, 2019; Willcox et al, 2012). There are three known γδ T cell subsets in humans that can be identified by the expression of the Vδ1, Vδ2 or Vδ3 chains, which will determine ligand recognition and TCR-driven effector functions (Willcox et al, 2020). Moreover, γδ T cells can be activated in the absence of TCR engagers (e.g. by cytokines) (Agerholm and Bekiaris, 2021), further emphasizing their potential in benefiting antigen mediated resistance of ICB therapy. The protective role of γδ T cells in cancer has been well documented in both mice and humans. This is attributed to their ability to become rapidly cytotoxic or cytokine-producing in an MHC-independent manner and home to relevant tissues (Mensurado et al, 2023). In fact, a large transcriptomic analysis across 18000 tumor samples identified intra-tumoral γδ T cells as the most significant favorable leukocyte population in the tumor microenvironment (Gentles et al, 2015). As such, to unravel the full potential of γδ T cells in the context of ICB therapy, it is important to elucidate the biology of ICRs in these cells and their response in patients after receiving ICB drugs. Although there are a number of mechanistic experiments in mice (Bekiaris et al, 2013; Edwards et al, 2023) as well as human studies (Catafal-Tardos et al, 2021) highlighting that ICRs can regulate the immunobiology of γδ T cells, these tend to focus on a single receptor or cell subset. This has created a paucity in our understanding of how ICRs work in γδ T cells and how to efficiently target them in ICB, especially when evidence shows that they can co-express a number of different ICRs (Brauneck et al, 2021).

Herein, we undertook a systematic study to address basic questions regarding the expression, regulatory role and clinical importance of ICB-relevant ICRs in human γδ T cells. We present evidence that TIGIT was restricted to Vδ1 whereas surface CTLA-4 was restricted to Vδ2 cells. TIM-3, LAG-3 and CTLA-4 were highly inducible by the TCR and interleukin(IL)-15, in a JAK/STAT dependent mechanism. PD-1 was expressed at similar levels and

could inhibit the function of both subsets. In melanoma, Vδ1 cells from patients but not healthy individuals displayed a transcriptional signature reminiscent of exhaustion characterized by genes coding ICRs and TOX, while all patient-derived γδ T cells had a marked downregulation of genes coding AP-1 transcription factors. Functionality in response to polyclonal stimuli was not affected by cancer or therapy, although induction of LAG-3 and CTLA-4 was impaired in both Vδ1 and Vδ2 cells. A second set of checkpoint receptors, namely the inhibitory killer Ig-like receptors (KIRs), were upregulated in Vδ1 cells from patients with melanoma and their expression was downregulated following successful ICB therapy. Finally, we found that in patients who responded to anti-PD-1 monotherapy, Vδ1 cells specifically bound high levels of therapeutic Ab, indicating that measuring Vδ1-bound anti-PD-1 could be used as a prognostic marker to evaluate the efficacy of ICB therapy.

# Results

## Differential expression of ICRs by γδ T cell subsets and regulation by cytokines and TCR

In order to examine the expression profile of some of the most well-studied ICRs in human γδ T cells we designed a flow cytometry panel so that we could simultaneously detect PD-1, TIGIT, TIM-3, LAG-3 and either surface(s) or total(t) (surface+intracellular) CTLA-4. We analyzed PBMCs from up to 22 healthy donors without prior manipulation and found that the levels of expression of most ICRs differed significantly between Vδ1 and Vδ2 cells (Fig. 1; Appendix Fig. S1A,B). Thus, whereas PD-1 levels were similar, TIGIT was almost exclusively and highly expressed by Vδ1 cells (Fig. 1A). The same pattern was true for TIM-3, although its expression levels were much lower (Fig. 1A). In contrast to TIGIT and TIM-3, LAG-3 was differentially expressed in Vδ2 cells, although its levels were very low (Fig. 1A). We could detect very little to no tCTLA-4 or sCTLA-4 (Fig. 1A). Hence, in the absence of stimuli, ICRs show preferential expression between the two γδ T cell subsets.

Next, we wanted to test whether cytokine or TCR stimulation could alter ICR expression levels and cultured total PBMCs, with either media alone, IL-2, IL-7, IL-15, or α-CD3. Since Vδ2 cells are reactive to phosphoantigens (pAg) through butyrophylin-TCR interactions (Willcox et al, 2023), we additionally stimulated PBMCs with HMBPP, a model pAg of the mevalonate pathway. In the absence of TCR stimulation, IL-15 had the most potent effect in inducing upregulation of ICRs (Appendix Fig. S1C–H), and we therefore focused our efforts on this cytokine. In Vδ1 cells, expression of PD-1 did not change after stimulation, with the exception of a small induction in the presence of both α-CD3 and IL-15 (Fig. 1B). We observed that in Vδ2, but not in Vδ1 cells, the level of PD-1 dropped after culture (Fig. 1A,B), however, it could be re-induced by IL-15 and TCR stimulation, while their combination had an additive effect (Fig. 1B). In contrast to PD-1, TIM-3 was readily induced by both IL-15 and TCR, while their combined signals induced the highest expression levels irrespective of cell subset (Fig. 1C). In response to TCR or TCR with IL-15, levels of TIM-3 were higher in Vδ2 than Vδ1 cells (Fig. 1C). Given the very high levels of TIGIT in Vδ1 cells, it was not surprising

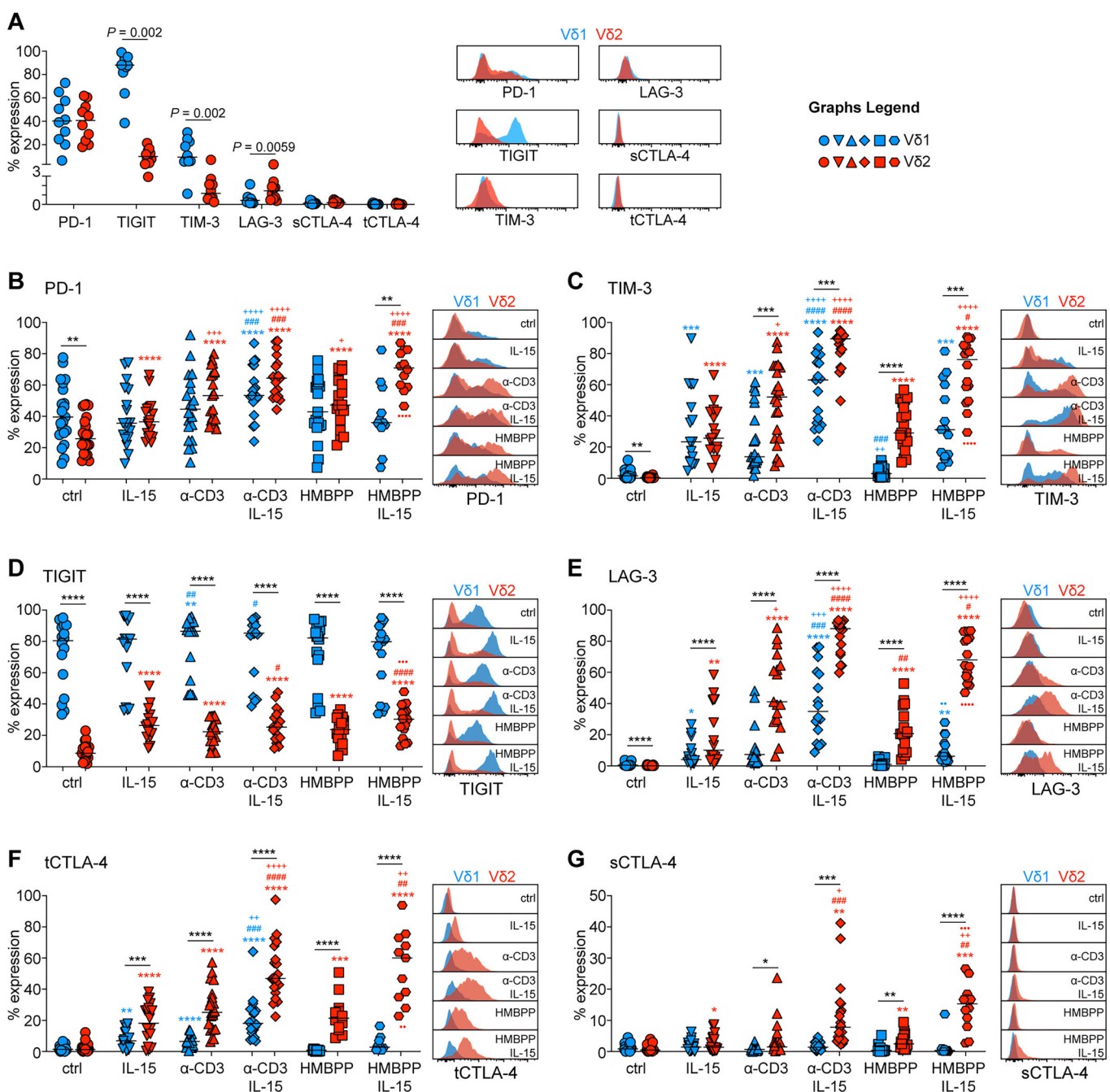

**Figure 1. Differential expression of ICRs by human γδ T cell subsets.**

Expression of PD-1 (**A, B**), TIM-3 (**A, C**), TIGIT (**A, D**), LAG-3 (**A, E**), total (t)CTLA-4 (**A, F**) and surface(s) CTLA-4 (**A, G**) in circulating Vδ1 (blue) and Vδ2 cells (red) from 10 healthy individuals (**A**). (**B–G**) PBMCs from 22 different healthy donors were stimulated for 48 h with IL-15, plate-bound α-CD3 antibody or HMBPP as indicated. *$P < 0.05$, **$P < 0.01$, ***$P < 0.001$ ****$P < 0.0001$ by paired Wilcoxon test (**A**) or mixed-effect model with the Geisser-Greenhouse correction and Tukey's multiple comparisons test (**B–G**). _ $P$ between Vδ1 and Vδ2. *$P$ versus control, +$P$ versus IL-15, #$P$ versus α-CD3, •$P$ versus HMBPP, in Vδ1 (blue) or red in Vδ2 (red); exact $P$ values can be found in Dataset EV2. Ctrl unstimulated control. Each symbol represents a donor. Source data are available online for this figure.

that it was largely unaffected by stimulation in this subset (Fig. 1D). However, TIGIT could be induced in Vδ2 cells by either IL-15 or TCR stimulation, but never to the same degree as in Vδ1 cells (Fig. 1D). LAG-3 was induced by both IL-15 and TCR stimulation with a clear synergistic effect, and with Vδ2 cells being significantly more responsive (Fig. 1E). tCTLA-4 levels were

induced in Vδ2 cells after stimulation, with the combination of TCR and IL-15 promoting the strongest expression (Fig. 1F). Vδ1 cells expressed significantly less tCTLA-4 irrespective of stimuli (Fig. 1F). sCTLA-4 was almost exclusively induced in Vδ2 cells, especially when TCR and IL-15 signals were combined (Fig. 1G).

In summary, these data reveal four ICR expression profiles between the two γδ T cell subsets. TIGIT is selectively and highly expressed by Vδ1 cells and is unresponsive to stimulation. PD-1 is equally and highly expressed by both subsets. TIM-3, LAG-3 and tCTLA-4 are only expressed after stimulation and their expression is always higher in Vδ2 cells. Finally, sCTLA-4 is selective to Vδ2 cells only after stimulation.

## The JAK/STAT pathway is required for TCR and IL-15 induced ICR expression in γδ T cells

IL-15 is a common γ-chain cytokine that upon binding to its receptor, it robustly induces downstream signaling through the JAK-STAT pathway. It has been recently shown that JAK inhibitors improve clinical responses and efficacy of anti-PD-1 therapy (Mathew et al, 2024; Zak et al, 2024). Furthermore, there is little molecular insight of how ICRs are regulated on the surface of human γδ T cells. Therefore, we wanted to test whether inhibition of STAT3/5 (JPX-0700) (Sorger et al, 2022), JAK1 (abrocitinib), JAK2 (NSC33994), JAK3 (low dose tofacitinib) or combined JAK1/2/3 (high dose tofacitinib) could suppress the induction of any of the ICRs in response to TCR (α-CD3) and IL-15 stimulation. We were also interested in potential subset-specific or stimulation-specific differences. Combined inhibition of JAK1/2/3 could suppress almost all ICRs, except TIGIT, on both subsets and had the strongest overall impact on reducing expression (Fig. EV1A,B; Appendix Fig. S2A–F). Similarly, STAT3/5 inhibition suppressed induction of most ICRs, including TIGIT in Vδ2 cells (Fig. EV1A,B; Appendix Fig. S2A–F). STAT3/5 inhibition failed to significantly reduce tCTLA-4 and sCTLA-4 expression on Vδ1 cells most likely reflecting the very low levels of CTLA-4 in this subset (Fig. EV1A; Appendix Fig. S2E,F). Targeting only JAK1 reduced expression of TIM-3, LAG-3 and tCTLA-4 in Vδ1 cells (Fig. EV1A; Appendix Fig. S2B,D,E), whereas in Vδ2 cells JAK1 additionally regulated induction of sCTLA-4 and PD-1 (Fig. EV1B; Appendix Fig. S2A,F). In Vδ1 cells, JAK2 inhibition affected only the expression of TIM-3 and only in the presence of IL-15, while in Vδ2 cells it also had a small but significant effect on LAG-3 and PD-1 (Fig. EV1A,B; Appendix Fig. S2A,B,D). Interestingly, single JAK2 inhibition had no impact on CTLA-4 levels but had the most profound impact on TIGIT expression and only in Vδ2 cells activated through the TCR and IL-15 (Fig. EV1B; Appendix Fig. S2C,E,F). JAK3 mildly regulated induction of all ICRs except TIGIT and tCTLA-4 but this was only in α-CD3 stimulated Vδ2 cells (Fig. EV1; Appendix Fig. S2A–F). In contrast, in Vδ1 cells JAK3 impacted only LAG-3 and sCTLA-4 and only when IL-15 was present (Fig. EV1A; Appendix Fig. S2D,F). Collectively, these data suggest that as well as being differentially expressed and induced in Vδ1 and Vδ2 cells, ICRs can also be differentially regulated by the JAK-STAT pathway.

## In vitro PD-1-mediated inhibition assays have minimal impact on human γδ T cell function

As PD-1 and its ligands are the most frequent targets of ICB therapy, we wanted to test whether it could inhibit Vδ1 and Vδ2 cells. To achieve this, we performed two independent assays. Hence, we magnetically depleted CD4⁺, CD19⁺ and CD14⁺ cells from PBMCs and cultured them with or without plate-bound recombinant PDL-1-Fc or PDL-2-Fc and one day later we assessed

the production of IFN-γ and TNF-α, as well as the expression of CD107a, which marks degranulating cells, by both γδ T cell subsets. However, in this experimental set up we did not observe any inhibition (Appendix Fig. S3A–C). Next, we instead co-cultured cells with irradiated K562 cell lines that constitutively expressed either PDL-1 or PDL-2 (Appendix Fig. S3D) in the presence of α-CD3 and IL-15. At day 2 after culture we assessed the production of IFN-γ, TNF-α and the expression of CD107a. We found that PDL-1 and PDL-2 could suppress IFN-γ production by both Vδ1 and Vδ2 cells, although TNF-α could be inhibited only in Vδ2 cells (Fig. EV2A–D). Furthermore, there was a small but significant inhibition of CD107a by PDL-1 in Vδ1 (Fig. EV2A,B) and by PDL-2 in Vδ2 cells (Fig. EV2C,D). We observed, however, that the function of Vδ1 cells was severely compromised upon culture with the K562 cell line even in the absence of PD-1 ligands (Appendix Fig. S3E). Vδ2 cells on the other hand were seemingly unaffected (Appendix Fig. S3F). Therefore, our data suggest that, at least in vitro, PD-1 signaling can inhibit the effector function of human γδ T cells, however this is not pronounced and appears dependent on the experimental system.

## Persistent binding of therapeutic antibody on Vδ1 cells discerns patients with melanoma who respond to α-PD-1 monotherapy

Next, we wanted to explore whether ICB therapy could impact the biology of γδ T cells and whether we could pinpoint putative therapeutic roles. We analyzed γδ T cells derived from PBMCs of patients with stage IV metastatic melanoma before starting either α-PD-1 monotherapy (pembrolizumab) or α-PD-1/α-CTLA-4 (nivolumab/ipilimumab) combination therapy. From the same patients PBMC samples were also obtained 3-4 months after treatment (Appendix Table S1; Fig. EV3A). Patients were stratified according to ICB treatment, and whether they responded or not (responders; R and non-responders; NR). In total, we analyzed 28 patients, 18 under monotherapy and 10 under combination therapy, of which approximately half in each group responded (Appendix Table S1; Fig. EV3A). Frequencies of γδ T cells as well as conventional CD4 and CD8 T cells were not affected by treatment or response and were similar to healthy controls, although we detected differences among T cell subsets (Fig. EV3B,C).

As one of our primary goals was to investigate the potential importance of ICRs in γδ T cell subsets in health and in cancer, we additionally assessed expression of PD-1, TIM-3, LAG-3, TIGIT and CTLA-4 by flow cytometry in all patients. We could not find any noticeable differences among patient groups or treatments or when compared to healthy controls and in general these data agreed with our results in Fig. 1 where we showed low expression of TIM-3, LAG-3 and CTLA-4 in the absence of stimulation and high levels of TIGIT in Vδ1 cells (Appendix Fig. S4). However, for every, but one patient group, Vδ1 cells had higher levels of total PD-1 than Vδ2 (Appendix Fig. S4A). In order to properly assess expression of PD-1, we had to take into account that therapeutic α-PD-1 can be detected on the surface of T cells long after treatment (Zelba et al, 2018). We therefore co-stained for PD-1 together with an α-IgG4 Ab corresponding to the isotype of the therapeutic α-PD-1, both pembrolizumab and nivolumab (Fig. 2A). Hence, we calculated total PD-1⁺ cells as the combined frequency

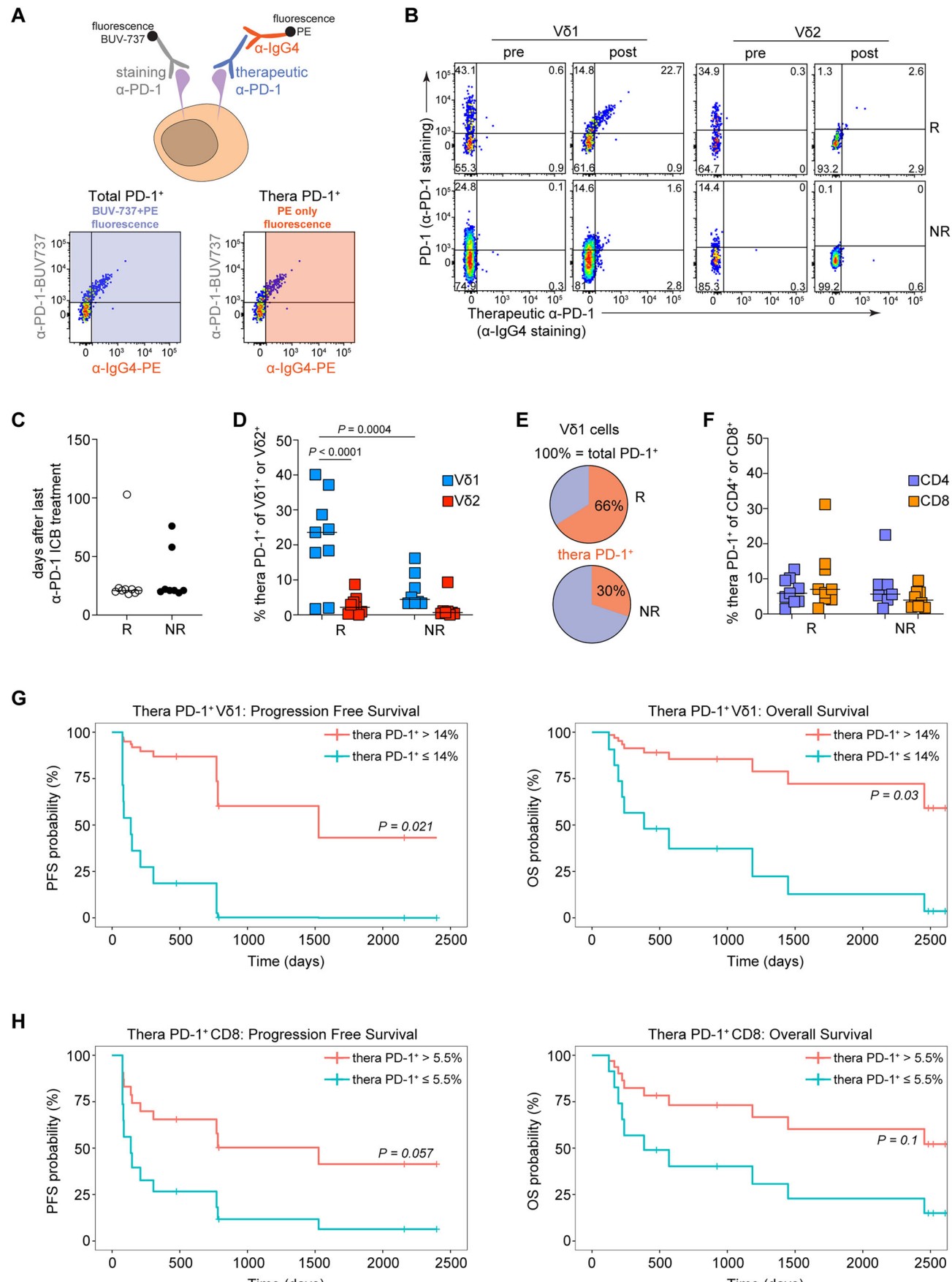

◀

**Figure 2. Binding of therapeutic antibody on Vδ1 cells discerns patients with melanoma who respond to α-PD-1 monotherapy.**

Analysis of PD-1 expression in PBMCs from patients with stage IV metastatic melanoma who either responded (R, $n = 9$) or did not respond (NR, $n = 8$) to pembrolizumab (therapeutic α-PD-1) treatment. Paired samples were obtained from these patients before (pre) and 3 to 4 months after (post) the start of immunotherapy. (A) To assess total PD-1 expression in these samples, we used two fluorescently labeled antibodies (left), targeting either PD-1 molecules (α-PD-1-BUV-737) or the therapeutic α-PD1 antibody (α-IgG4-PE). The total frequency of PD-1$^+$ cells was determined as cells that were positive for α-PD-1-BUV-737 and α-IgG4-PE; the frequency of therapeutic antibody binding (thera PD-1$^+$) was determined as cells that were positive only for α-IgG4-PE. (B) Representative flow cytometric analysis of therapeutic α-PD-1 antibody binding on the surface of Vδ1 and Vδ2 cells. (C) Days between the last antibody infusion and the collection of samples. Each symbol represents a patient. (D) Thera PD-1$^+$ Vδ1 and Vδ2 cells after pembrolizumab treatment. $P$ values were calculated by paired two-way ANOVA and Tukey's multiple comparisons test. Each symbol represents a patient. (E) Frequency of Vδ1 cells with bound therapeutic antibody out of the total PD-1$^+$ Vδ1 population after pembrolizumab treatment. The pie charts show average expression in 9 R and 8 NR patients. (F) Thera PD-1$^+$ CD4 and CD8 T cells after pembrolizumab treatment. Each symbol represents a patient. (G) Progression-free survival (PFS; $\beta = -2.5$, $P = 0.021$, HR $= 0.083$ with 95% CI of 0.01–0.68), and overall survival (OS; $\beta = -1.8$, $P = 0.03$, HR $= 0.16$ with 95% CI of 0.03–0.84) curves according to Cox proportional hazard model of patients with melanoma based on frequency of thera PD-1$^+$ Vδ1 (cut-off >14%). (H) Progression-free survival (PFS; $\beta = -1.1$, $P = 0.057$, HR $= 0.32$ with 95% CI of 0.099–1), and overall survival (OS; $\beta = -1.1$, $P = 0.1$, HR $= 0.34$ with 95% CI of 0.095–1.2) curves according to Cox proportional hazard model of patients with melanoma based on frequency of thera PD-1$^+$ CD8 (cut-off >5.5%). (G, H) The cut-off was calculated as the average between the median frequency of thera PD-1$^+$ cells in patients who responded to therapy and the median frequency of thera PD-1$^+$ cells in the patients who did not respond to therapy. Source data are available online for this figure.

of cells positive for fluorescent signals derived both from α-IgG4 and our α-PD-1 staining Ab (see Appendix Fig. S4A as example of total PD-1), while therapeutic α-PD-1$^+$ cells (thera PD-1$^+$) were calculated as the fluorescence signal coming only from α-IgG4 (Fig. 2A). Therapeutic α-PD-1 could be easily detected in patients who responded to monotherapy, but almost exclusively on Vδ1 cells (Fig. 2B). Patients who did not respond to monotherapy had very low levels of detectable therapeutic Ab (Fig. 2B). This was not due to differences in the timing of treatment (Fig. 2C). When we quantified thera PD-1$^+$ cells we found that they were significantly enriched only in the Vδ1 population from patients who responded (Fig. 2D,E). Binding of therapeutic α-PD-1 on conventional CD4 and CD8 T cells was lower and could not discern responder patients (Fig. 2F). To assess the association of therapeutic α-PD-1 binding with patient survival, we used Cox hazard regression models. We observed that therapeutic α-PD-1 binding to Vδ1 ($\beta = -0.065$, $P = 0.044$, HR $= 0.94$ with 95% CI of 0.88–1) and CD8 ($\beta = -0.24$, $P = 0.036$, HR $= 0.78$ with 95% CI of 0.62–0.98) T cells was associated with significantly longer progression-free survival in patients with melanoma. Next, we stratified patients based on frequencies of therapeutic α-PD-1 binding to Vδ1 and CD8 T cells. Patients with high frequencies of circulating thera PD-1$^+$ Vδ1 cells ( > 14%; calculated as the average between the median frequency of thera PD-1$^+$ cells in patients who responded to therapy and the median frequency of thera PD-1$^+$ cells in the patients who did not respond to therapy) but not thera PD-1$^+$ CD8 ( > 5.5%; calculated as for Vδ1 cells) showed significantly longer progression-free survival and overall survival compared with patients with low therapeutic antibody binding (Fig. 2G,H).

In patients that were treated with combination therapy, thera-PD-1$^+$ γδ T cells were also constrained within the Vδ1 subset, however, there was no difference between responder and non-responder groups (Fig. EV4A–D). Notably, in these patients, binding of therapeutic α-PD-1 on Vδ1 cells was at very high levels, surpassing levels on CD4 or CD8 T cells and what we observed in monotherapy treated patients (Fig. EV4C,D and compare to Fig. 2D–F). Collectively, these data suggest that persistent binding to and detection of pembrolizumab on Vδ1 γδ T cells may be useful in predicting the patients who will respond to monotherapy.

## Transcriptional profiling reveals functional properties of γδ T cell subsets and a cancer-specific loss of AP-1 transcription factors

The above data prompted us to further investigate the effects of ICB therapy on γδ T cell biology. Hence, we purified total blood γδ T cells from the above patient groups and from healthy controls and performed scRNA-seq, a technique that is widely used to assess the transcriptional state of multiple cell types in an unbiased manner. We used Uniform Manifold Approximation and Projection (UMAP) analysis to visualize varying γδ T cell subsets and by integrating all samples and all conditions, including healthy controls, we identified 4 clusters which contained all the major subsets of Vδ1, Vδ2 and Vδ3 cells (Fig. 3A; Appendix Fig. S5). Cluster 0 (c0) was almost exclusively Vδ2, while the majority of c1 were Vδ1 cells (Fig. 3A). Vδ3 cells were interspersed within the Vδ1 clusters (Fig. 3A), while c2 and c3 contained a mix of all populations (Fig. 3A). To get a better grasp about the identity of the clusters, we investigated the expression of some of the top immunologically related genes that showed cluster-specificity. Noticeable differences between c0 and c1 were the high expression of *KLRC1*, *IL7R*, *GZMK* and *THEMIS* in c0 and the high expression of *TIGIT*, *KIR2DL3*, *KIR3DL1*, *KIR3DL2* and *KLRC3* in c1 (Fig. 3B), most likely reflecting differences between Vδ1 and Vδ2 cells. Cluster c1 additionally expressed high levels of the classical NK cell marker *NCR1*, which agrees with the well-established relationship between γδ T and NK cells in both human and mouse (Haas et al, 2009; Pizzolato et al, 2019). C2 expressed a mix of c0 and c1 genes as well as a group of genes associated with transcriptionally active cells (e.g. *MED13L*, *ANK3*, *KLF12*) (Fig. 3B). C3 expressed genes related to lymph node (LN) homing and naïve T cells such as *CCR7*, *SELL*, *S1PR1* and *NELL2* as well as the transcription factors *LEF1* and *TCF7*, which are important for T cell development (Fig. 3B). The distribution and abundance of these clusters did not change among the 24 patients with melanoma that we analyzed and was not affected by treatment (Fig. EV5A). Since γδ T cells can also be categorized as type 1 or type 3 we examined the expression of *IFNG* and *IL17A*. We found *IFNG* being expressed in all clusters (Fig. EV5B) but we could not detect any *IL17A* or *IL17F*, and fewer than 10 cells with transcripts for *IL17C* and *IL17D*. However, we could detect *RORC* and *IL23R* in cluster c0 (Fig. EV5B).

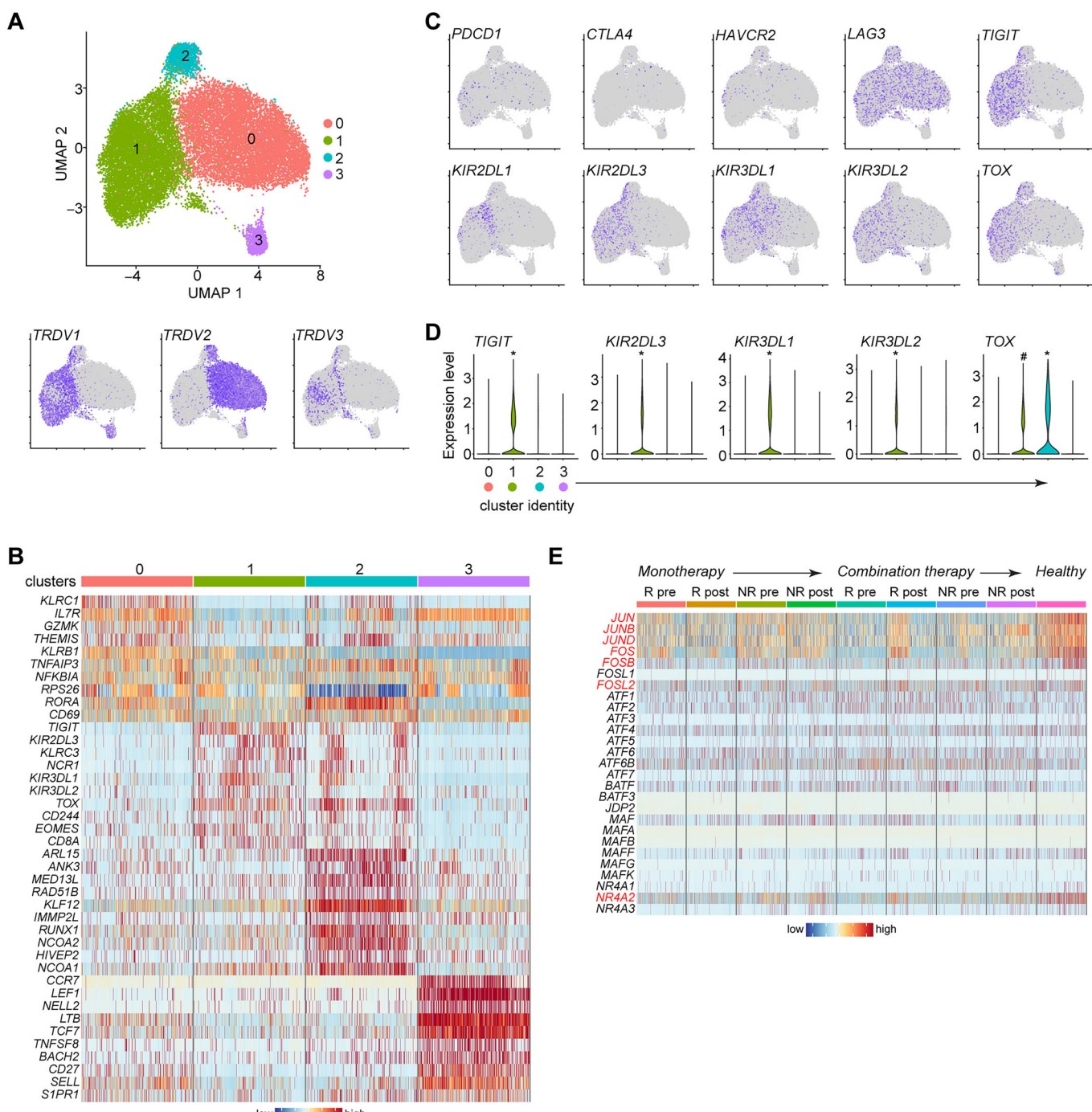

**Figure 3. Transcriptomic characterization of melanoma derived γδ T cells shows subset-specific characteristics and regulation of AP-1 transcription factors.**

(A) UMAP embedding showing the clustering of circulating γδ T cells ($n = 84{,}537$) derived from healthy donors ($n = 8$) and patients with stage IV melanoma who received α-PD-1 (monotherapy, $n = 16$) or α-PD-1 and α-CTLA-4 (combination therapy, $n = 8$) treatment; and distribution of *TRDV1*, *TRDV2*, *TRDV3* among these clusters. (B) Heatmap showing the normalized expression of 10 of the most differentially expressed genes (DEGs) in each cluster. (C) Distribution of *PDCD1*, *CTLA4*, *HAVCR2*, *LAG3*, *TIGIT*, *KIR2DL1*, *KIR2DL3*, *KIR3DL1*, *KIR3DL2* and *TOX* expression in this dataset. (D) Expression of *TIGIT*, *KIR2DL3*, *KIR3DL1*, *KIR3DL2* and *TOX* in each cluster. (E) Heatmap showing expression of AP-1 and NR4A transcription factors in γδ T cells in the different melanoma patient groups and healthy donors. R, responder; NR, non-responder; pre, samples were taken before the start of treatment; post, samples were taken 3 to 4 months after the start of treatment. Cluster 0, $n = 45676$; cluster 1, $n = 37{,}871$; cluster 2, $n = 3411$; cluster 3, $n = 1962$. DEGs were identified by Wilcoxon Rank Sum test as those with an average Log2 fold change of at least 0.5 and an adjusted $P < 0.05$. In violin plots, * indicates a statistically significant upregulation in this cluster against all other clusters, and # indicates significant upregulation in cluster c1 against clusters c0 and c3 (adjusted $P < 0.0001$); exact $P$ values can be found in Dataset EV1.

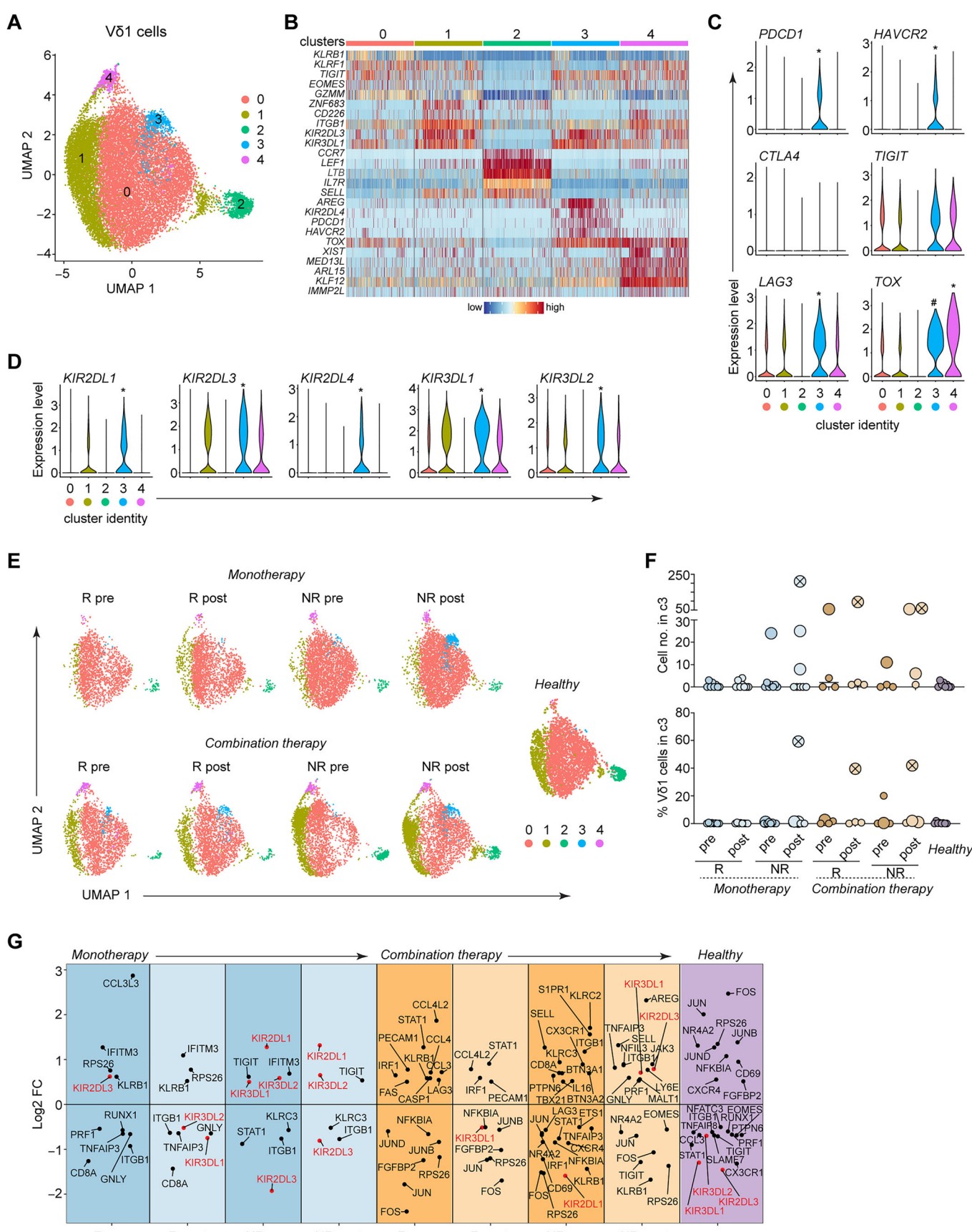

◀ **Figure 4.  Transcriptomic characterization of Vδ1 cells indicates association between ICR and KIR transcripts with cancer status and ICB responses.**

(A) UMAP embedding showing the clustering of cells expressing *TRDV1* (*n* = 27,827) derived from healthy donors (*n* = 8) and patients with stage IV melanoma who received α-PD-1 monotherapy (*n* = 16) or α-PD-1 and α-CTLA-4 combination therapy (*n* = 8). (B) Heatmap showing the normalized expression of 5 of the most differentially expressed genes (DEGs) in each cluster. (C) Expression of *PDCD1, LAG3, HAVCR2, TIGIT, CTLA4* and *TOX* in each cluster. (D) Expression of *KIR2DL1, KIR2DL3, KIR2DL4, KIR3DL1* and *KIR3DL2* in each cluster. (E) UMAP embedding of *TRDV1*-expressing cells in each patient group. (F) Number of cells in cluster c3 (top) and frequency of *TRDV1*-expressing cells in c3 (bottom) in each sample. Each dot represents a patient sample; larger dots represent samples with 6 or more cells in c3; crossed circles represent samples with high frequency of cells in cluster 3. (G) Differential expression of immune-relevant genes between patient groups. DEGs were identified by Wilcoxon Rank Sum test as those with an average Log2 fold change (FC) of at least 0.5 and an adjusted *P* < 0.05. In violin plots, * indicates a statistically significant upregulation in this cluster against all other clusters, and # indicates significant upregulation in cluster c3 against clusters c0, c1 and c2 (adjusted *P* < 0.0001); exact *P* values can be found in Dataset EV1. R responder, NR non-responder, pre samples were taken before the start of treatment, post samples were taken 3 to 4 months after the start of treatment. Cluster 0, *n* = 20,635 (74% of total); cluster 1, *n* = 5227 (18.8% of total); cluster 2, *n* = 916 (3.3% of total); cluster 3, *n* = 584 (2.1% of total); cluster 4, *n* = 465 (1.7% of total).

We then examined expression of the ICRs that we have been studying and found that transcripts for *PDCD1* (PD-1), which were unexpectedly low, were similarly distributed among the subsets, while *CTLA4* was mostly associated with the Vδ2 dominant c0 (Fig. 3C), agreeing with the protein data in Fig. 1. *HAVCR2* (TIM-3) levels were also low, reflecting its low steady-state protein expression (Fig. 1A,C), and did not have a particular distribution pattern (Fig. 3C). *LAG3* showed robust expression except in c3. *TIGIT* was restricted in the Vδ1 dominant c1 and in c2 (Fig. 3C) and its expression was significantly higher in c1 (Fig. 3D). We also investigated the expression of inhibitory KIRs and found that they were mostly associated with the Vδ1 dominant c1 (Figs. 3C and EV5D), while *KIR2DL3, KIR3DL1* and *KIR3DL2* were significantly higher in this cluster (Fig. 3D). *TOX*, which is often used to mark exhausted cells, was also most highly expressed in clusters c1 and c2 (Fig. 3C,D). These data indicate that at the transcriptional level, Vδ1 cells are characterized by high expression of TIGIT, TOX and inhibitory KIRs.

We additionally examined differentially expressed genes (DEGs) among all patient groups for γδ T cells and identified a number of immunologically relevant genes that were being modulated, however, not at any clear observable pattern that reflected disease status or type of therapy (Dataset EV1), most likely due to the mixed γδ T cell populations in this dataset. Nevertheless, when we considered all the top non-curated genes that were significantly different among patient groups, we noted that *JUN, FOS, FOSB*, three AP-1 transcription factors and *NR4A2* were highly expressed in healthy but not patient derived γδ T cells (Fig. EV5C). Hence, we investigated the expression levels of all AP-1 and NR4A family members and found that *JUN, JUNB, JUND, FOS, FOSB* and *FOSL2*, together with *NR4A2* were always downregulated in all patient groups (Fig. 3E). Collectively, these data show that although the circulating pool of γδ T cells was relatively unperturbed in melanoma, their transcriptional profile underwent specific changes that correlated with disease.

## ICR and KIR transcripts in Vδ1 cells are associated with cancer status and ICB responses

To get a better understanding of transcriptional changes among different γδ T cell subsets, we re-clustered the cells based on their expression of *TRDV1, TRDV2* and *TRDV3* (Figs. 4, 5 and 6). Within the Vδ1 population we found 5 clusters, and 4 clusters each for Vδ2 and Vδ3 cells (Figs. 4A, 5A and 6A). Although there was clear heterogeneity in cluster abundance among patient groups before and after therapy, which may affect the overall outcome of

therapeutic efficiency, the distribution and abundance of these clusters was not statistically significant (Appendix Fig. S6A–C). To define the nature of each cluster we examined the top DEGs and found that similar to the general γδ T cell population, every subset had a cluster with genes related to active transcription and a cluster with genes related to naïve and LN homing T cells (Figs. 4B, 5B and 6B). The other clusters were separated based on genes corresponding to effector functions, such as KIRs in Vδ1, granzymes in Vδ2 and killer cell like receptors in Vδ3 cells (Figs. 4B, 5B and 6B).

A closer look at Vδ1 cells revealed that ICRs and KIRs were prominent among the genes that separated the clusters (Fig. 4B). In particular, cluster c3 expressed significantly more *PDCD1, HAVCR2* and *LAG3* than any other cluster, and had very high levels of *TOX* (Fig. 4C), while it also expressed the highest levels of *KIR2DL1, KIR2DL3, KIR2DL4, KIR3DL1* and *KIR3DL2* (Fig. 4D). Interestingly, c3 was barely present in healthy individuals, but was more prominent in patients with melanoma, especially those who required combination therapy and those who did not respond to monotherapy (Fig. 4E,F), suggesting that melanoma favors the appearance of Vδ1 cells with a potentially exhaustive transcriptional make up. Admittedly, this cluster consisted of few cells (Fig. 4F, top) and corresponded only to 2.1% of the total *TRDV1*-expressing population (Fig. 4), and as such we cannot speculate on its overall contribution to disease state. It is plausible, however, that this cluster of cells may have patient-specific significance, since in 3 of the contributing patients it represented a large proportion of the total Vδ1 population (Fig. 4F, bottom).

Next, we investigated DEGs in all Vδ1 cells among the different patient groups. Many of the genes that were modulated in each patient group were similar to the changes we observed for the total γδ T cell population (Dataset EV1) (e.g. the AP-1 signature in healthy controls versus patients). However, we observed that the high expression of KIR transcripts in Vδ1 cells was associated mostly with cancer patients and not healthy controls (Fig. 4G). Moreover, KIR gene expression was downregulated in those patients that underwent successful ICB therapy (Fig. 4G). For example, *KIR3DL1* and *KIR3DL2* were significantly downregulated in monotherapy responders after ICB, but were upregulated together with *KIR2DL1* in patients who did not respond to monotherapy and their expression was not reduced after ICB (Fig. 4G). Similarly, *KIR3DL1* was reduced in patients who responded to combination therapy but was highly expressed in patients who did not (Fig. 4G). In Vδ2 and Vδ3 cells, expression of ICRs was low, with the exception of *LAG3* and *TIGIT* (Vδ3 only), and with no clear distribution patterns among the clusters (Figs. 5C and 6C). In this regard, we could not identify a cluster reminiscent

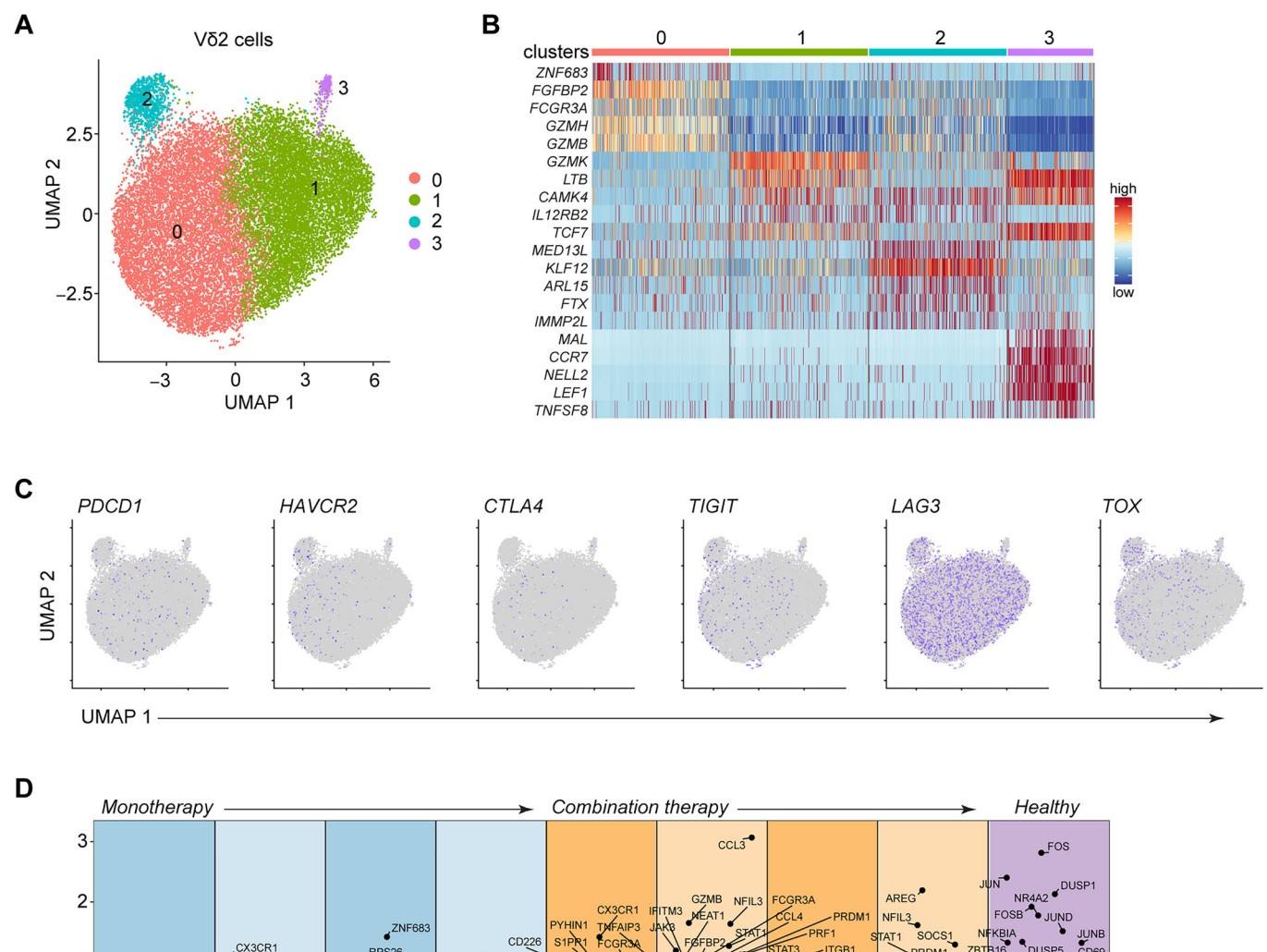

**Figure 5. Transcriptomic characterization of Vδ2 cells.**

(A) UMAP embedding showing the clustering of cells expressing *TRDV2* ($n = 42,814$) derived from healthy donors ($n = 8$) and patients with stage IV melanoma who received α-PD-1 monotherapy ($n = 16$) or α-PD-1 and α-CTLA-4 combination therapy ($n = 8$). (B) Heatmap showing the normalized expression of 5 of the most differentially expressed genes (DEGs) in each cluster. (C) Expression of *PDCD1*, *LAG3*, *HAVCR2*, *TIGIT*, *CTLA4* and *TOX* in each cluster. (D) Differential expression of immune-relevant genes between patient groups. R responder, NR non-responder, pre samples were taken before the start of treatment, post samples were taken 3 to 4 months after the start of treatment. Cluster 0, $n = 22326$; cluster 1, $n = 19,222$; cluster 2, $n = 1017$; cluster 3, $n = 249$. DEGs were identified by Wilcoxon Rank Sum test as those with an average Log2 fold change (FC) of at least 0.5 and an adjusted $P < 0.05$.

of Vδ1 c3. Furthermore, although there were many immunologically relevant DEGs that were modulated in the patient groups, we did not observe any specific expression patterns of specific or groups of genes (Figs. 5D and 6D). Together, these data suggest that modulation of KIR gene expression in Vδ1 cells may be able to provide clues regarding the outcome of ICB therapy.

## γδ T cells from patients with melanoma are functional following polyclonal stimulation but show impaired induction of LAG-3 and CTLA-4

Despite the downregulation of AP-1 and NR4A transcription factors in γδ T cells derived from patients or the apparent

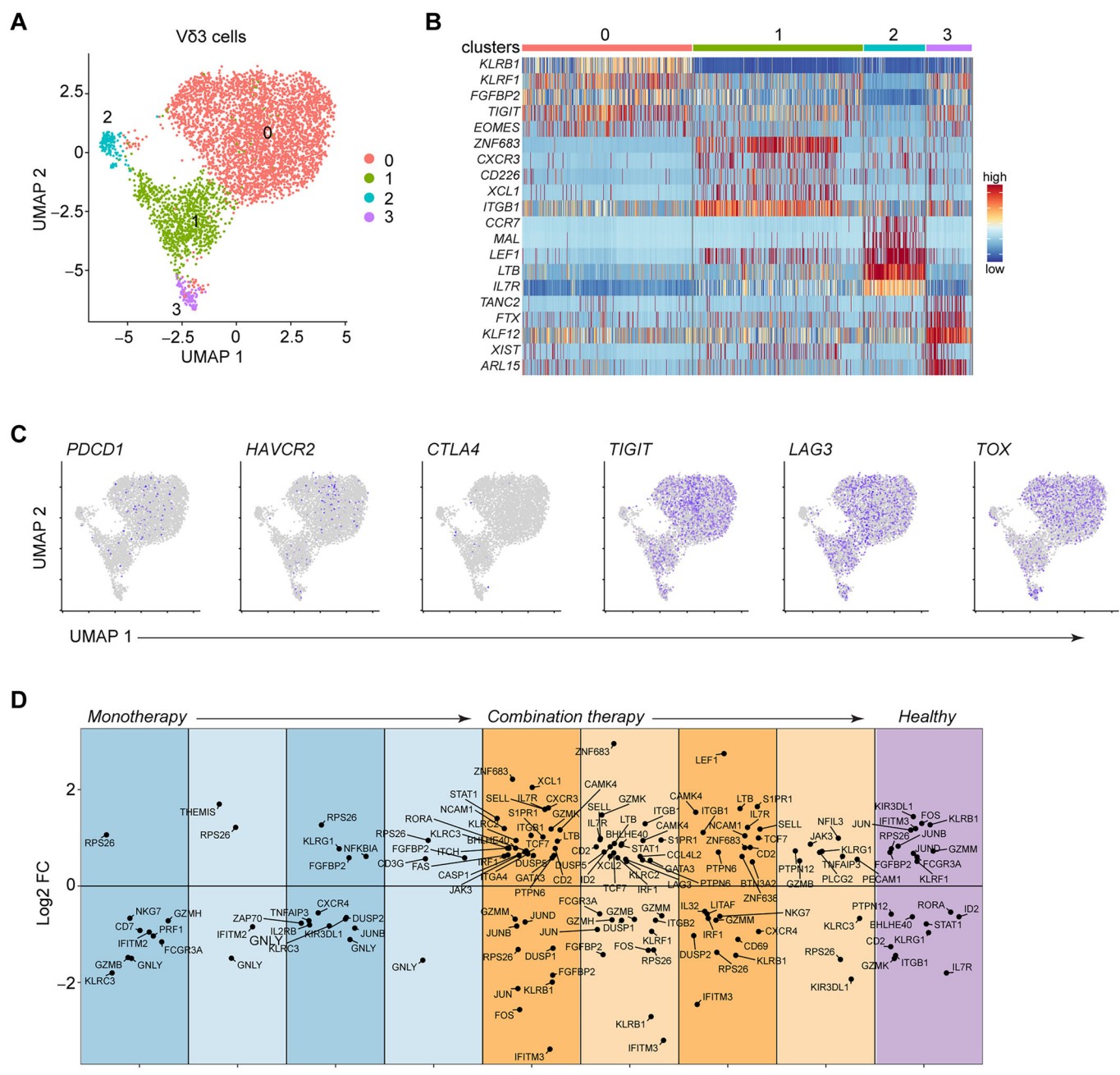

**Figure 6. Transcriptomic characterization of Vδ3 cells.**

(A) UMAP embedding showing the clustering of cells expressing *TRDV3* ($n = 4937$) derived from healthy donors ($n = 8$) and patients with stage IV melanoma who received α-PD-1 monotherapy ($n = 16$) or α-PD-1 and α-CTLA-4 combination therapy ($n = 8$). (B) Heatmap showing the normalized expression of 5 of the most differentially expressed genes (DEGs) in each cluster. (C) Expression of *PDCD1*, *LAG3*, *HAVCR2*, *TIGIT*, *CTLA4* and *TOX* in each cluster. (D) Differential expression of immune-relevant genes between patient groups. R responder, NR non-responder, pre samples were taken before the start of treatment, post samples were taken 3 to 4 months after the start of treatment. Cluster 0, $n = 3657$; cluster 1, $n = 1029$; cluster 2, $n = 144$; cluster 3, $n = 107$. DEGs were identified by Wilcoxon Rank Sum test as those with an average Log2 fold change (FC) of at least 0.5 and an adjusted $P < 0.05$.

expression of exhaustion-associated transcripts, neither Vδ1 nor Vδ2 cells showed impaired functionality following polyclonal activation with PMA and ionomycin (Appendix Figs. S7 and S8), besides reduced frequencies of IFN-γ single producing Vδ2 cells in patients prior to receiving α-PD-1 monotherapy (Appendix

Fig. S7E). In contrast, frequencies of polyfunctional cells (positive for two or three of the measured functional parameters IFN-γ, TNF-α, CD107a) were often higher in patients compared to healthy controls (Appendix Figs. S7A,C,D and S8A). Therefore, under these conditions, neither cancer nor ICB treatment status appears to

negatively affect the functionality of circulating γδ T cells. Although impaired γδ T cell function has been reported before in melanoma (Argentati et al, 2003), our data agree with more recent findings which have suggested that despite transcriptional signatures reminiscent of exhaustion, γδ T cells in cancer patients can retain their function (Davies et al, 2024; Rancan et al, 2023).

Next, we cultured healthy and patient PBMCs with or without α-CD3 and IL-15 for two days to promote induction of ICRs (similar to Fig. 1B–G). We compartmentalized our analysis based on γδ T cell subset (Vδ1 or Vδ2) and on therapy (monotherapy or combination) and compared them to the same healthy controls. Furthermore, as we have shown earlier that α-CD3 + IL-15 induces robust expression of most ICRs (Fig. 1B–G), we do not present or discuss statistical differences between control and activated cells. However, all reported statistical differences were calculated by intercalating the total variance of control and activated cells.

In Vδ1 cells, induction and expression of PD-1, TIGIT and TIM-3 was not affected by disease state or therapy (Appendix Fig. S9A–H). However, induction of LAG-3 and tCTLA-4 was significantly impaired in non-responder patients before starting monotherapy with ICB not having a clear impact (Fig. 7A,B,D). sCTLA-4 could only be visibly induced in healthy Vδ1 cells, however, this was not significant (Fig. 7C,D). In Vδ1 cells from patients who received combination therapy, LAG-3 induction remained low and did not recover after ICB (Fig. 7E,H), tCTLA-4 was unchanged (Fig. 7F,H), while sCTLA-4 could only be detected after treatment and in healthy controls, albeit at very low levels (Fig. 7G,H).

In Vδ2 cells, induction of PD-1 and TIM-3 was slightly higher but insignificant in healthy controls compared to patients irrespective of ICB therapy, while TIGIT levels were unchanged (Appendix Fig. S10A–H). Similar to Vδ1 cells, induction of LAG-3 in the Vδ2 subset was significantly impaired and its levels did not recover after ICB therapy (Fig. 8A,D,E,H). Induction of tCTLA-4 was also compromised and its expression levels were not restored after therapy (Fig. 8B,D,F,H). Noticeably, sCTLA-4 failed to be induced and rescued in Vδ2 cells from patients with melanoma irrespective of their condition or what kind of ICB therapy they received (Fig. 8C,D,G,H). Collectively, these data show that in melanoma, although γδ T cells appear capable of producing cytokines, induction of LAG-3 and CTLA-4 in response to TCR and IL-15 is impaired.

## Discussion

The present study comprises a comprehensive attempt to study the expression and regulation of ICRs in human γδ T cells and to elucidate the importance of these cells in cancer patients before and after ICB therapy. Hence, γδ T cells displayed subset-specific expression and regulation of ICRs, such as constitutive Vδ1-restricted TIGIT or inducible Vδ2-restricted sCTLA-4. Induction of ICRs was modulated by combined signals from cytokines and the TCR through the JAK-STAT pathway. To a certain extent, γδ T cells were also prone to ICR inhibition. Circulating γδ T cells from patients with cancer could be distinguished from healthy by downregulation of a group of AP-1 transcription factor genes. A minor subset of Vδ1 cells detected by scRNA-seq showed an exhaustion transcriptional signature, mainly of ICR and KIR coding

genes, that was prominent in certain patients who did not respond to ICB therapy. Polyclonal ex vivo stimulation did not reveal any obvious functional defects in any of the γδ subsets. However, TCR and IL-15 mediated induction of LAG-3 and CTLA-4 was impaired in both Vδ1 and Vδ2 cells derived from patients. In the context of ICB therapy, we provide evidence that Vδ1 cells only from patients who responded to α-PD-1 monotherapy uniquely bound high levels of therapeutic α-PD-1. Finally, inhibitory KIR gene expression by Vδ1 cells was correlated with cancer and was reduced after successful ICB therapy.

Although γδ T cells have taken center stage in numerous critical immunological functions, including cancer and cancer immunotherapy, their association with ICRs has been somewhat neglected. Thus, we know little regarding mechanisms that regulate ICR expression in γδ T cells. For example, a combination of TCR stimuli (e.g. pAg or zoledronate) and IL-2 can induce surface expression of PD-1 in Vδ2 cells (Guo et al, 2020; Hoeres et al, 2019; Iwasaki et al, 2011). TIM-3 can also be induced under similar conditions (Guo et al, 2020), while IL-21 could upregulate TIM-3 in Vδ2 cells in the absence of TCR stimuli (Wu et al, 2019). Our data show that in circulating γδ T cells, TCR and cytokine stimulation, especially IL-15 through the JAK-STAT pathway, are potent inducers of TIM-3, LAG-3 and CTLA-4, suggesting that these ICRs are rapidly inducible and may be reliable markers of recently activated γδ T cells. PD-1 was constitutively expressed and could only be marginally modulated by exogenous stimuli. TIGIT was constitutive and at very high levels in Vδ1 cells only and refractory to further stimulation. However, as Vδ1 cells are largely tissue-resident we cannot rule out that the TIGIT expression we observed was induced in the tissue before the cells exited into the blood. Interestingly, CTLA-4 levels were much higher in Vδ2 cells, which almost exclusively expressed sCTLA-4. It is plausible that CTLA-4 can directly suppress Vδ2 cells or their surface expression of it may turn them into suppressor cells by transendocytosing CD80/86 from antigen presenting cells. Collectively, the differential expression and regulation of ICRs in γδ T cell subsets may have important implications in elucidating mechanisms of ICB therapy.

Recent work has demonstrated that in the context of cancer, despite displaying a transcriptional signature reminiscent of exhausted T cells, γδ T cells do not appear to become functionally exhausted (Davies et al, 2024; Rancan et al, 2023). In our melanoma cohort, scRNA-seq analysis showed the presence of a Vδ1 subpopulation displaying an exhaustion transcriptional signature characterized primarily by high levels of transcripts for ICRs, KIRs and TOX. This was especially evident in patients who did not respond to therapy and those who were qualified for α-PD-1/α-CTLA-4 combination therapy, but not in healthy controls. Although this subpopulation made up on average only about 2% of the total Vδ1 compartment, in 3 out the 24 patients it represented 40-60% of all Vδ1 cells, suggesting a putative patient-specific importance. Irrespective of subsets, we also found that all γδ T cells from patients with melanoma and independent of ICB outcome had a significant reduction in the expression of a number of genes coding AP-1 transcription factors, which are important in augmenting TCR and co-stimulation driven T cell activation (Atsaves et al, 2019). In line with these findings, a seminal study by Ahmed and Wherry showed that a feature of exhausted CD8 T cells in the context of chronic viral infection is the downregulation of AP-1 transcription factors (Wherry et al, 2007). However, despite

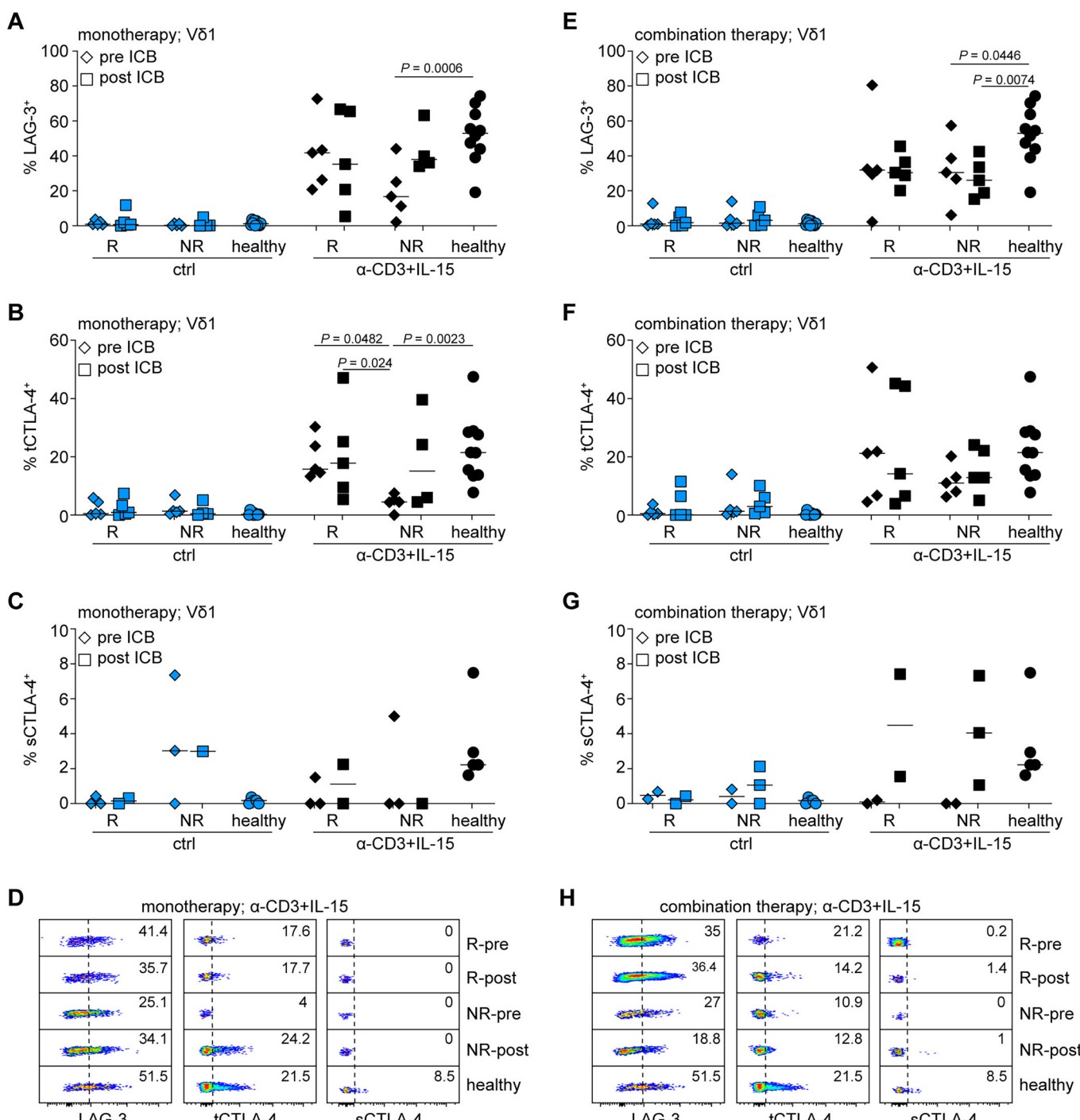

**Figure 7. ICR induction in Vδ1 cells from patients undergoing immunotherapy.**

Analysis of Vδ1 cells from patients with stage IV melanoma who were treated with either α-PD1 alone (monotherapy, (A–D) or α-PD-1 and α-CTLA-4 (combination therapy; (E–H)). Samples were obtained from patients who responded (R) or did not respond (NR) to treatment both before (pre) and 3 to 4 months after (post) the start of immunotherapy. PBMCs were stimulated with α-CD3 and IL-15 for 48 h before assessment of LAG-3 (A, E), total (t)CTLA-4 (B, F) and surface (s)CTLA-4 (C, G) expression in Vδ1 cells. (D, H) Representative flow cytometry analysis in these samples. P values were calculated by paired two-way ANOVA and Tukey's multiple comparisons test. In graphs, each symbol represents a donor. Patient samples included in the analysis can be found in Fig. EV3A. Healthy donors (A, B, E, F) n = 10; (C, G) n = 5. Source data are available online for this figure.

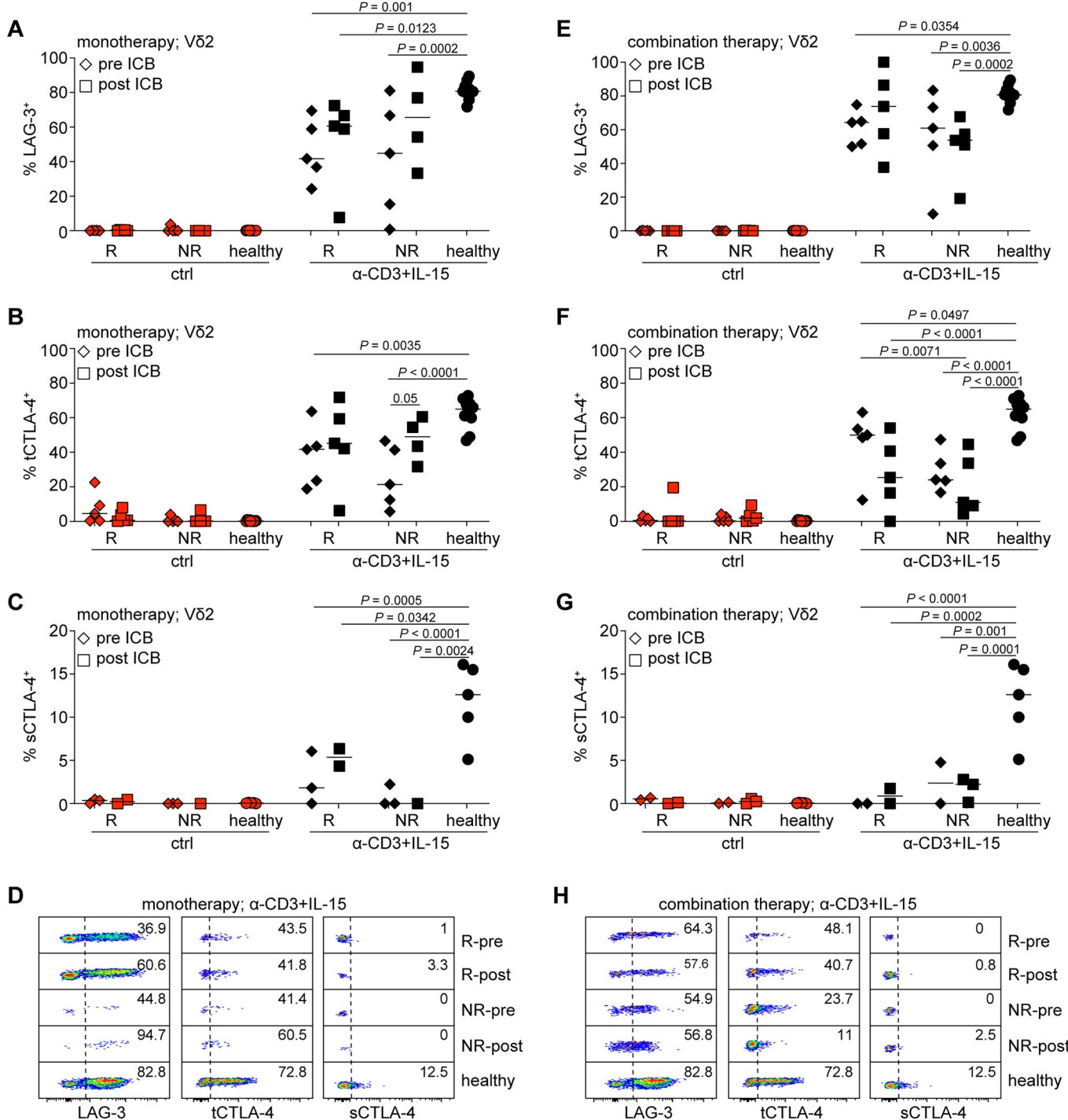

**Figure 8. ICR induction in Vδ2 cells from patients undergoing immunotherapy.**

Analysis of Vδ2 cells from patients with stage IV melanoma who were treated with either α-PD1 alone (monotherapy, (**A–D**) or α-PD-1 and α-CTLA-4 (combination therapy; (**E–H**)). Samples were obtained from patients who responded (R) or did not respond (NR) to treatment both before (pre) and 3 to 4 months after (post) the start of immunotherapy. PBMCs were stimulated with α-CD3 and IL-15 for 48 h before assessment of LAG-3 (**A, E**), total (t)CTLA-4 (**B, F**) and surface (s)CTLA-4 (**C, G**) expression in Vδ2 cells. (**D, H**) Representative flow cytometry analysis in these samples. *P* values were calculated by paired two-way ANOVA and Tukey's multiple comparisons test. In graphs, each symbol represents a donor. Patient samples included in the analysis can be found in Fig. EV3A. Healthy donors (**A, B, E, F**) *n* = 10; (**C, G**) *n* = 5. Source data are available online for this figure.

the presence of transcriptional "signs" indicating that γδ T cells will be exhausted, direct ex vivo re-activation, albeit with strong polyclonal stimuli, did not show any functional impairment. In fact, γδ T cells from patients with melanoma showed polyfunctionality with robust production of IFN-γ, TNF-α and expression of CD107a, which agrees with previous reports (Davies et al, 2024; Rancan et al, 2023). The only compromised response that we could detect was that patient-derived Vδ1 and Vδ2 cells had impaired induction of LAG-3 and CTLA-4 upon TCR and IL-15 stimulation. It is important to acknowledge that although the above data represent circulating cells in the blood, tumor infiltrating γδ T cells may have different functional and transcriptional profiles but also different levels of expression of ICRs. However, tumor samples are rarely retrieved from patients who respond to ICB therapy, and thus comparative analyses of responder and non-responder patient groups like the one presented herein would not be possible.

Almost 10 years ago, Gentles et al reported that tumor infiltration by γδ T cells is the most significant parameter to predict a favorable anti-cancer response (Gentles et al, 2015). Since numerous studies have provided significant evidence that a favorable prognosis among various cancers is associated with strong cytotoxic type 1 γδ T cell responses, especially by the Vδ1 subset (Mensurado et al, 2023). Our data reveal a Vδ1 property that may be useful in predicting and evaluating the response to α-PD-1 monotherapy. Thus, compared to other circulating T cell populations, Vδ1 cells had high levels of therapeutic α-PD-1 on their surface and this was restricted to patients who responded positively to therapy. In parallel with these results, a recent study showed that a Merkel cell carcinoma patient who completely responded to pembrolizumab had high levels of PD-1 in both tumor infiltrating and circulating Vδ1 cells (Lien et al, 2024). Interestingly, in patients that received combination therapy, binding of therapeutic α-PD-1 to Vδ1 cells was almost 3 times higher and still more than other T cell subsets, however, this could not distinguish responder from non-responder patients. The increased binding in these patients, which was observed across T cell populations, could not be attributed to more recent treatments, and it is unlikely to be due to the different α-PD-1 clinical products that the patients received (pembrolizumab and nivolumab) as they both have similar half-life (Centanni et al, 2019). Although this will have to be evaluated with larger patient cohorts, we would like to suggest that a relatively simple blood test focused on α-PD-1-bound Vδ1 could be a valuable means of evaluating α-PD-1 monotherapy.

In addition to the ICRs that have been discussed herein, a second family of checkpoint receptors have emerged as potential targets for cancer immunotherapy. These are the inhibitory KIRs which are primarily expressed by natural killer (NK) cells and deliver a negative signal by binding to HLA molecules (Djaoud and Parham, 2020; Pende et al, 2019). Early studies with a blocking Ab against KIR2DL1-3 showed improved NK cell cytotoxicity (Benson et al, 2011; Romagne et al, 2009) and currently there are anti-KIR therapeutic antibodies in clinical trials (highlighted at https://www.innate-pharma.com). Mechanistically, inhibitory KIRs contain intracellular ITIMs and ITSMs that have been shown to inhibit NK cell function through the recruitment of SHP-1 and SHP-2 (Ren et al, 2022; Wei et al, 2021). KIRs have been described to be expressed by human γδ T cells many years ago (Fisch et al, 1997), and it was recently shown that tumor infiltrating Vδ1 cells can express high levels of KIR transcripts (de Vries et al, 2023).

Interestingly, KIR3DL3 was found to be preferentially expressed in tissue resident γδ T cells and to suppress TCR mediated activation (Palmer et al, 2023). Our scRNA-seq data showed that transcripts for inhibitory KIRs were specifically enriched in Vδ1 cells, especially in the subpopulation that also displayed high co-expression of ICR genes and *TOX*. KIR gene expression correlated mostly with cells derived from patients with melanoma and it was downregulated following successful mono- or combination therapy. Whether increased expression of KIRs is a feature of γδ T cells in some malignancies is not known. It is plausible, however, that high KIR expression by Vδ1 cells is the result of an anti-tumor response. On the other hand, the reduced expression of KIRs following successful ICB therapy may reflect the overall positive immune response to the cancer.

In summary, our data provide a comprehensive and systematic analysis of ICRs in human γδ T cells at the level of expression, regulation and cancer and highlight that these evolutionary conserved lymphocytes can provide us with unique tools to discern the success and outcome of standard care immunotherapy.

# Methods

**Reagents and tools table**

| Reagent/resource | Reference or source | Identifier or catalog number |
| --- | --- | --- |
| **Experimental models** | | |
| PDL-1⁻ PDL-2⁻ K562 cells (*H. sapiens*) | Dr. Kristoffer Haurum Johansen, DTU, Denmark | Dr. Kristoffer Haurum Johansen, DTU, Denmark |
| PDL-1⁺ K562 cells (*H. sapiens*) | Dr. Kristoffer Haurum Johansen, DTU, Denmark | Dr. Kristoffer Haurum Johansen, DTU, Denmark |
| PDL-2⁺ K562 cells (*H. sapiens*) | Dr. Kristoffer Haurum Johansen, DTU, Denmark | Dr. Kristoffer Haurum Johansen, DTU, Denmark |
| **Recombinant DNA** | | |
| **Antibodies** | | |
| Ultra-LEAF purified anti-human CD3 | Biolegend | 317326 |
| Anti-human CD3 FITC | Biolegend | 317306 |
| Anti-human TCRγδ BV421 | BD Bioscience | 744870 |
| Anti-human TCRVδ1 PE-Cy7 | ThermoFisher | 25-5679-42 |
| Anti-human TCRVδ1 APC | ThermoFisher | 17-5679-42 |
| Anti-human TCRVδ2 PE-Dazzel594 | Biolegend | 746567 |
| Anti-human TCRVδ2 BV480 | BD Bioscience | 331426 |
| Anti-human CD4 BV510 | BD Bioscience | 562970 |
| Anti-human CD4 BUV737 | BD Bioscience | 568369 |
| Anti-human CD8 PerCp-Cy5.5 | BD Bioscience | 560662 |
| Anti-human CTLA-4 PE-Cy7 | Biolegend | 369614 |

| Reagent/resource | Reference or source | Identifier or catalog number |
| --- | --- | --- |
| Anti-human CTLA-4 APC-Cy7 | Biolegend | 369634 |
| Anti-human PD-1 BUV737 | BD Bioscience | 612791 |
| Anti-human IgG4 PE | Southern Biotech | 9200-09 |
| Anti-human LAG-3 BV786 | BD Bioscience | 744727 |
| Anti-human TIM-3 BV650 | BD Bioscience | 565584 |
| Anti-human TIGIT PE | BD Bioscience | 568672 |
| Anti-human TIGIT AF647 | BD Bioscience | 568944 |
| Anti-human CD107a BV650 | Biolegend | 328638 |
| Anti-human IFN-γ BUV395 | BD Bioscience | 563563 |
| Anti-human IFN-γ APC | BD Bioscience | 554702 |
| Anti-human TNF-α APC-Cy7 | Biolegend | 502944 |
| Anti-human TNF-α BV605 | BD Bioscience | 569308 |
| Anti-human IL-17A BV786 | BD Bioscience | 563745 |
| Anti-human CD4 Biotin | Biolegend | 300504 |
| Anti-human TCRαβ Biotin | Biolegend | 306704 |
| Anti-human CD19 Biotin | Biolegend | 302204 |
| Anti-human CD14 Biotin | Biolegend | 301826 |
| TotalSeq™-C0251 anti-human Hashtag 1 Antibody | Biolegend | 394661 |
| TotalSeq™-C0252 anti-human Hashtag 2 Antibody | Biolegend | 394663 |
| TotalSeq™-C0253 anti-human Hashtag 3 Antibody | Biolegend | 394665 |
| TotalSeq™-C0254 anti-human Hashtag 4 Antibody | Biolegend | 394667 |
| **Oligonucleotides and other sequence-based reagents** | | |
| **Chemicals, enzymes and other reagents** | | |
| Dulbecco's phosphate-buffered saline (PBS) | Gibco | 14190-250 |
| Lymphoprep™ solution | STEMCELL Technologies | 18060 |
| Heat-inactivated fetal bovine serum (FBS) | Gibco | A5256801 |
| DMSO | Sigma-Aldrich | D2650 |
| HEPES | Gibco | 15630-056 |
| 2-mercaptoethanol | Gibco | 31350-010 |
| L-glutamine | Gibco | 25030-024 |
| Penicillin Streptomycin | Gibco | 15140-122 |
| (E)-4-hydroxy-3-methylbut-2-enyl pyrophosphate (HMBPP) | Cayman Chemical | 154-13580-500 |
| Recombinant human IL-15 | Biotechne | 247-ILB-005/CF |
| Recombinant human IL-7 | Biotechne | 207-IL-010/CF |
| Recombinant human IL-2 | Biolegend | 589104 |
| Abrocitinib | Sigma-Aldrich | 1622902-68-4 |
| NSC33994 | Sigma-Aldrich | 82058-16-0 |
| Tofacitinib citrate | Sigma-Aldrich | 540737-29-9 |

| Reagent/resource | Reference or source | Identifier or catalog number |
| --- | --- | --- |
| JPX-0700 | H. Sorger et al (2022) | Prof. Dr. Richard Moriggl, VUW, Austria |
| Ionomycin | Sigma-Aldrich | I0634 |
| Phorbol myristate acetate (PMA) | Sigma-Aldrich | P1585 |
| BD GolgiStop™ | BD Bioscience | 554724 |
| BD Horizon™ Fixable Viability Stain 700 | BD Bioscience | 564997 |
| BD Cytofix/Cytoperm™ Fixation/Permeabilization Kit | BD Bioscience | 554714 |
| EasySep RaphidSphere streptavidin beads | STEMCELL technologies | 50001 |
| PDL-1-Fc recombinant protein | Biolegend | 762506 |
| PDL-2-Fc recombinant protein | Biolegend | 772506 |
| Human TruStainFcX™ | Biolegend | 422302 |
| Bovine serum albumin (BSA) | Sigma | A2153 |
| Chromium Next GEM Single Cell 5′ Kit v2 | 10X Genomics | PN-1000265 |
| Chromium 5′ Feature Barcode Kit | 10X Genomics | PN-1000541 |
| Dual Index Kit TT Set A | 10X Genomics | PN-1000215 |
| Dual Index Kit TN Set A | 10X Genomics | PN-1000250 |
| High Sensitivity DNA kit | Aligent | 5067-4626 |
| KAPA Library Quantification Kit for Illumina Platforms | Kapa Biosystems | KK4824 |
| **Software** | | |
| FlowJo version | BD Bioscience | v10.10.0 |
| bcl2fastq | Illumina | v2.20 |
| CellRanger | 10X Genomics | v7.1.0 |
| Seurat | Satija Lab | v5.0.1 |
| Prism | GraphPad | v10.2.3 |
| Survival package | Comprehensive R Archive Network (CRAN) | v3.8-3 |
| **Other** | | |
| BD LSRFortessa | BD Bioscience | |
| BD FACSAria Fusion | BD Bioscience | |
| EasySep™ Magnet | STEMCELL technologies | 18000 |
| Chromium Next GEM Chip K | 10X Genomics | PN-1000287 |
| Chromium Single Cell Controller | 10X Genomics | PN-120263 |
| Illumina NextSeq 2000 | Illumina | |
| Illumina NovaSeq 6000 | Illumina | |

## Methods and protocols

### PBMC isolation

Healthy donor samples were obtained from the Central Blood Bank at Rigshospitalet (Copenhagen, DK) under approval from the local Ethics Committee. Peripheral blood mononuclear cells (PBMCs) were isolated from buffy coats by density gradient centrifugation. Blood samples were diluted 1:1 with Dulbecco's phosphate-buffered saline (PBS, Gibco) and were distributed into conical tubes containing Lymphoprep™ solution (STEMCELL technologies). These tubes were centrifuged at 800 RCF for 25 min at room temperature and with decreased deceleration. The cells from the interphase were then collected, washed three times with PBS (Gibco), and cryopreserved in heat-inactivated fetal bovine serum (FBS, Gibco) with 10% DMSO (Sigma).

### Patient samples

Cryopreserved PBMCs from patients with stage IV metastatic melanoma were obtained from the National Center for Cancer Immune Therapy (CCIT-DK) at Copenhagen University Hospital, Herlev. All enrolled patients provided oral and written informed consent before blood sampling. The study was approved by the local Ethics Committee (H-15007985) and experiments conformed to the principles set out in the WMA Declaration of Helsinki and the Department of Health and Human Services Belmont Report. The patient cohort consisted of individuals that responded (R) or did not respond (NR) to four series of Pembrolizumab (mono; R, $n = 10$; NR, $n = 8$), or a combination of Nivolumab and Ipilimumab (combi; R, $n = 5$; NR, $n = 5$). Eight of the patients did not receive the full course of treatment due to toxicity (mono R, $n = 2$; mono NR, $n = 1$; combi R, $n = 2$; combi NR, $n = 3$). The age and sex distribution of the participants in each group can be found in Appendix Table S1. Samples were collected from the participants at baseline and around three months after the start of therapy. Patients that responded to treatment were defined as those with an objective response (RECIST v1.1 criteria) or progression-free survival for longer than 365 days.

### Cell culture and cell stimulation

Cryopreserved PBMCs were thawed and maintained in culture media containing RPMI 1640 (Invitrogen) supplemented with 10% heat-inactivated FBS (Gibco), 20 mM HEPES (pH 7.4; Gibco), 50 μM 2-mercaptoethanol, 2 mM L-glutamine (Gibco), and 100 U/mL penicillin–streptomycin (Gibco). For ICR quantification, $2 \times 10^6$ total PBMC were cultured for 2 days at 37 °C 5% $CO_2$ in 24-well plates under several stimulatory conditions, as stated in the figures. To induce global TCR stimulation, culture plates were coated with 1 μg/ml α-CD3 antibody (OKT-3). To induce activation of Vγ9Vδ2 cells, 5 ng/ml of (E)-4-hydroxy-3-methylbut-2-enyl pyrophosphate (HMBPP, Cayman Chemical) were added to the media. For cytokine stimulation, 10 ng/ml of recombinant human(h) IL-15 (Biotechne), 20 ng/ml of hIL-7 (Biotechne), or 100 IU of hIL-2 (Biolegend) were added to the media. JAK1 inhibition was achieved with abrocitinib 200 nM (Sigma-Aldrich); JAK2 inhibition with NSC33994 1 μM (Sigma-Aldrich); and tofacitinib citrate (Sigma-Aldrich) was used at 10 nM to inhibit JAK3 signaling and at a high concentration of 1 μM for global JAK1/2/3 inhibition. For STAT3/5 inhibition, JPX-0700 was used at 1 μM (Sorger et al, 2022). For cytokine quantification, the cells were treated with ionomycin (750 ng/ml; Sigma), phorbol myristate acetate (PMA, 50 ng/ml; Sigma), and BD GolgiStop (1 μl/ml; containing monensin) for 3.5 h before harvesting, unless otherwise stated, at 37 °C and 5% $CO_2$.

### Flow cytometry staining

PBMCs were stained with Fixable Viability Stain 700 (1:1000, BD Horizon) in PBS for 10 min on ice. Surface antigens were stained in fluorescent-activated cell sorting (FACS) buffer prepared with PBS 3% FBS. Pan-TCRγδ was first stained for 20 min on ice, followed by all additional surface antigens except for CTLA-4, which were incubated for 30 min on ice. In order to evaluate both surface and total (surface + intracellular) CTLA-4, the samples were split into two. Surface CTLA-4 staining was performed on ice for 30 min. In order to stain for intracellular antigens, the cells were fixed and permeabilized by incubation in BD fixation/permeabilization solution for 20 min at room temperature, followed by washing twice in BD Perm/Wash buffer. Intracellular cytokines and total CTLA-4 were subsequently stained in BD Perm/Wash buffer for 30 min on ice. Lastly, the cells were washed twice with BD Perm/Wash buffer and acquired using BD LSRFortessa. The data was analyzed using FlowJo version 10.10.0 software (BD Biosciences). The flow cytometry antibodies used in this study are: α-CD3 (OKT3; FITC; 1:50), α-TCRγδ (11F2; BV421; 1:50), α-TCRVδ1 (TS8.2; PE-Cy7; 1:100), α-TCRVδ1 (TS8.2; APC; 1:20), α-TCRVδ2 (B6; PE-Dazzel594; 1:50), α-TCRVδ2 (B6; BV480; 1:200), α-CD4 (SK3; BV510; 1:50), α-CD4 (SK3; BUV737; 1:100), α-CD8 (RPA-T8; PerCP-Cy5.5; 1:50), α-CTLA-4 (BIN3; PE-Cy7; 1:20), α-CTLA-4 (BIN3; APC-Cy7; 1:20), α-PD-1 (EH12.1; BUV737; 1:20), α-IgG4 (HP6025; PE; 1:40), α-LAG-3 (T47-530; BV786; 1:20), α-TIM-3 (7D3; BV650; 1:100), α-TIGIT (TgMab-2; PE; 1:40), α-TIGIT (TgMab-2; AF647; 1:20), α-CD107a (H4A3; BV650; 1:20), α-IFN-γ (B27; BUV395; 1:100), α-IFN-γ (B27; APC; 1:200), α-TNF-α (MAb11; BV605; 1:20), α-TNF-α (MAb11; APC-Cy7; 1:20), α-IL-17A (N49-653; BV786; 1:20),

### Detection of therapeutic α-PD-1 antibodies

Pembrolizumab and Nivolumab are IgG4 antibodies that bind to PD-1 receptors. To detect therapeutic antibody bound on the surface of T cells, PBMCs were stained with α-IgG4 (1:40; HP6025) for 30 min on ice prior to staining of other surface markers. To detect total PD-1 expression in patient-derived PBMCs after culture, the harvested cells were first incubated in 10 μg/ml of Pembrolizumab in PBS for 30 min on ice, and were then stained for α-IgG4 and other markers as described.

### PD-1 functional assays

In order to enrich for γδ T cells, PBMCs were labeled in FACS buffer with biotinylated α-CD4 (SK3; 1:100), α-TCRαβ (IP26; 1:100), α-CD19 (HIB19; 1:100), and α-CD14 (HCD14; 1:100) for 15 min at room temperature. Next, 75 μl/ml EasySep RaphidSphere streptavidin beads were added to the cell suspensions. These samples were incubated for 5 min at room temperature and were transferred to an EasySep magnet for an additional 5 min. The cells remaining in the supernatant were used in two different assays. To evaluate the response to PD-1 ligands, α-CD3 antibody (1 μg/ml; OKT-3) and recombinant PDL-1-Fc and PDL-2-Fc chimera proteins (10 μg/ml; Biolegend) were coated on a 96-well tissue culture plate for 2 h. The wells were then washed twice with PBS. After enrichment, the T cells were resuspended in culture media containing 10 ng/ml of hIL-15 and were cultured on the α-CD3/ PDL-1-Fc/ PDL-2-Fc coated plate overnight at 37 °C and 5% $CO_2$. Alternatively, K562 cell lines that stably express PDL-1, PDL-2 or no ligand were kindly provided by Dr. Kristoffer Haurum

Johansen, DTU, Denmark. Ligand expression was verified by flow cytometry and PCR confirmed no mycoplasma contamination. The cell lines were irradiated with 40 Gy, and co-cultured with the enriched cell suspension. The cultures were plated on wells coated with α-CD3 antibody (1 μg/ml; OKT-3) at a T:K562 ratio of 2:1, and were cultured for two days at 37 °C and 5% $CO_2$. At day one, half of the media was discarded and substituted with fresh culture media containing IL-15. For cytokine quantification, PMA (50 ng/ml; Sigma), ionomycin (750 ng/ml; Sigma), and BD GolgiStop (BD Bioscience, 1 μl/ml) were added to all cultures 5 h before harvesting.

### Single-cell RNA-sequencing

γδ T cells from 16 patients that received Pembrolizumab, 8 patients that received Nivolumab+Ipilimumab, and 8 healthy controls were FACS-sorted for scRNA-seq. Cryopreserved PBMCs were thawed, washed and resuspended in Human TruStainFcX (1:10, Biolegend) diluted in PBS with 0,5% bovine serum albumin (BSA) for 10 min on ice. Next, 1 μl of a TotalSeq anti-human Hashtag Antibody (C0251, C0252, C0253, C0254; Biolegend) was added to each sample to be able to later pool them in groups of four. Thus, paired samples from one patient that responded to treatment, and paired samples from one patient that did not respond to treatment each received a distinct hashing antibody. Healthy controls were stained in a similar manner. After a 30-minute incubation, the cells were also stained with Fixable Viability Stain 700, α-CD3, α-CD4 and α-TCRγδ, and γδ T cells were then sorted as CD3$^+$CD4$^-$TCRγδ$^+$ using a BD FACSAria Fusion. After sorting, pooled γδ T cells were adjusted to the desired concentration, and were prepared according to 10X Genomics guidelines (Chromium Next GEM Single Cell 5' v2 (Dual Index) User Guide, CG000331 Rev E). Thus, samples were loaded on a Chromium Next GEM Chip K (PN-1000287) and run on a Chromium Single Cell Controller (PN-120263). scRNA-seq libraries were prepared according to manufacturer's instructions using Chromium Next GEM Single Cell 5' Kit v2 (PN-1000265), Chromium 5' Feature Barcode Kit (PN-1000541), Dual Index Kit TT Set A (PN-1000215), and Dual Index Kit TN Set A (PN-1000250). Library quality control and quantification was performed using the 2100 Bioanalyzer equipped with a High Sensitivity DNA kit (Agilent, 5067-4626), and the KAPA Library Quantification Kit for Illumina Platforms (Kapa Biosystems, KK4824). Equimolar amounts of each library were sequenced on an Illumina NextSeq 2000 instrument at the Genomics Core Facility "KFB - Center of Excellence for Fluorescent Bioanalytics" (University of Regensburg, Regensburg, Germany) or an Illumina NovaSeq 6000 instrument at the Flow Cytometry & Single Cell Core Facility at the University of Copenhagen, Denmark. The resulting .cbcl files were converted into .fastq files with the bcl2fastq v2.20 software.

### Initial processing of single-cell RNA-sequencing data

Sequencing reads were aligned to the reference genome GRCh38 version 2020 and processed for unique molecular identifier (UMI) barcodes using CellRanger v7.1.0. Further quality control and processing of the samples was performed using Seurat v5.0.1 (Hao et al, 2024). Cells with less than 500 genes and cells with more than 4000 genes were excluded from the dataset. In addition, low quality cells with more than 7% of mitochondrial counts, and cells expressing hemoglobin genes were discarded. HTODemux was used to assign cells to their sample of origin, and only cells identified as singlets were kept for further analysis. The individual samples were integrated with the IntegrateLayers function using the

harmony method (Korsunsky et al, 2019). The effects of cell cycle genes were regressed out when scaling the gene expression. During dimensionality reduction, the first 40 principal components were used to identify cell clusters, and the data was visualized with Uniform Manifold Approximation and Projection (UMAP). After initial clustering, contaminating cells were removed. The cells excluded expressed either FCN1, VCAN and MS4A6A, common in monocytes and macrophages; TRAV and TRAJ genes; or increased expression of cell cycle genes. Cells with no detectable expression of TRDV genes were also removed from the dataset. In addition to the global analysis, cells were separated based on expression of single TRDV genes. Thus, cells with transcripts of either TRDV1 (Vδ1 cells), TRDV2 (Vδ2 cells), or TRDV3 genes (Vδ3 cells) were separated, re-clustered, and analyzed independently.

### Gene expression analysis

Seurat's FindAllMarkers function was used to identify differentially expressed genes (DEGs) in our dataset with Wilcoxon Rank Sum test and an average log2 fold change of at least 0.5. The statistically significant genes (adjusted $P < 0.05$) were selected for further analysis. The top DEGs between clusters were illustrated with the DoHeatmap function, and due to the different sizes of the clusters, we decided to show gene expression for a maximum of 400 cells per cluster in each heatmap. The FeaturePlot function was used to visualize the expression of selected genes. The distribution of cells in the different clusters was calculated in samples with a minimum of 50 cells as the frequency of cells in a sample that were assigned to each cluster.

### Statistical analysis

Statistical analysis was performed in GraphPad Prism software version 10.2.3 unless otherwise stated and was not randomized or

**The paper explained**

**Problem**

Immune checkpoint blockade (ICB) therapies targeting the PD-1 receptor are widely used to treat various cancers. Although ICB therapies indiscriminately target all cells that express the immune checkpoint receptor (ICR), most studies have focused on conventional CD4 and CD8 T cells. γδ T cells are polyfunctional lymphocytes that can be exceptionally effective at killing cancers and are therefore potential targets of ICB. However, γδ T cells have only recently begun to be systematically studied in relation to ICRs and ICB.

**Results**

We identified constitutive and inducible ICR expression patterns in human γδ T cell subsets, then studied their response to anti-PD-1-based immunotherapy. We analyzed circulating γδ T cells from patients with melanoma and demonstrated that different subsets exhibited distinct transcriptional and functional characteristics associated with cancer and clinical responses to ICB. We found that, in patients who successfully responded to anti-PD-1 therapy, Vδ1 γδ T cells specifically bound high levels of therapeutic antibody.

**Impact**

The differential expression of ICRs by γδ T cell subsets could provide insight into the types of responses that will be elicited by different ICB therapies. Furthermore, the specific biological properties of γδ T cells, such as the persistent binding of the anti-PD-1 therapeutic, could be useful in predicting the efficacy of ICB therapy for individual patients.

blinded. Cox proportional hazards regression model was fitted using the survival package version 3.8-3. The statistical tests used in each figure are stated in the legends.

## Data availability

The sequencing data generated in this study has been deposited at the European Genome-phenome Archive (EGA) under accession number EGAS50000001270. The code used for reproducing analysis and figures is available at https://github.com/elcatar/gdT_melanoma (https://doi.org/10.5281/zenodo.15722184).

The source data of this paper are collected in the following database record: biostudies:S-SCDT-10_1038-S44321-025-00338-9.

## Peer review information

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

## Acknowledgements

We thank Dr. Kristoffer Haurum Johansen (DTU, Denmark) for kindly providing the K562 cell lines used in this study. This work was supported by funding provided to V.B. from the Danish Cancer Society (Kræftens Bekæmpelse, grant number R269-A15747).

## Author contributions

**Elisa Catafal-Tardos**: Conceptualization; Data curation; Software; Formal analysis; Validation; Investigation; Visualization; Methodology; Writing—original draft; Project administration; Writing—review and editing. **Lola Dachicourt**: Formal analysis; Validation; Investigation. **Maria Virginia Baglioni**: Investigation; Writing—review and editing. **Marcelo Gregorio Filho Fares da Silva**: Investigation; Writing—review and editing. **Davide Secci**: Investigation; Writing—review and editing. **Marco Donia**: Resources; Writing—review and editing. **Anders Handrup Kverneland**: Resources; Writing—review and editing. **Inge Marie Svane**: Resources; Writing—review and editing. **Vasileios Bekiaris**: Conceptualization; Resources; Formal analysis; Supervision; Funding

acquisition; Investigation; Visualization; Methodology; Writing—original draft; Project administration; Writing—review and editing.

Source data underlying figure panels in this paper may have individual authorship assigned. Where available, figure panel/source data authorship is listed in the following database record: biostudies:S-SCDT-10_1038-S44321-025-00338-9.

## Disclosure and competing interests statement

The authors declare no competing interests.

# Expanded View Figures

**Figure EV1.  Differential regulation of ICRs in human γδ T cell subsets.**

Flow cytometry analysis of PD-1, TIM-3, TIGIT, LAG-3, total (t)CTLA-4 and surface (s)CTLA-4 in Vδ1 (**A**) and Vδ2 (**B**) cells. PMBCs were stimulated for 48 h with α-CD3 or α-CD3 and IL-15 or left untreated (ctrl) in the presence of inhibitors as follows: abrocitinib was used for JAK1 inhibition; NSC33994 for JAK2 inhibition; JPX-0700 for STAT3/5 dual inhibition; and tofacitinib citrate was used at a concertation 10 nM to inhibit JAK3 signaling, and at a concentration of 1 μM for combined JAK1/2/3 inhibition. The heatmaps show average expression in cells from 13 different donors in four independent experiments. Detailed statistics for all possible comparisons in accompanying Appendix Fig. S2.

**A** Vδ1 cells

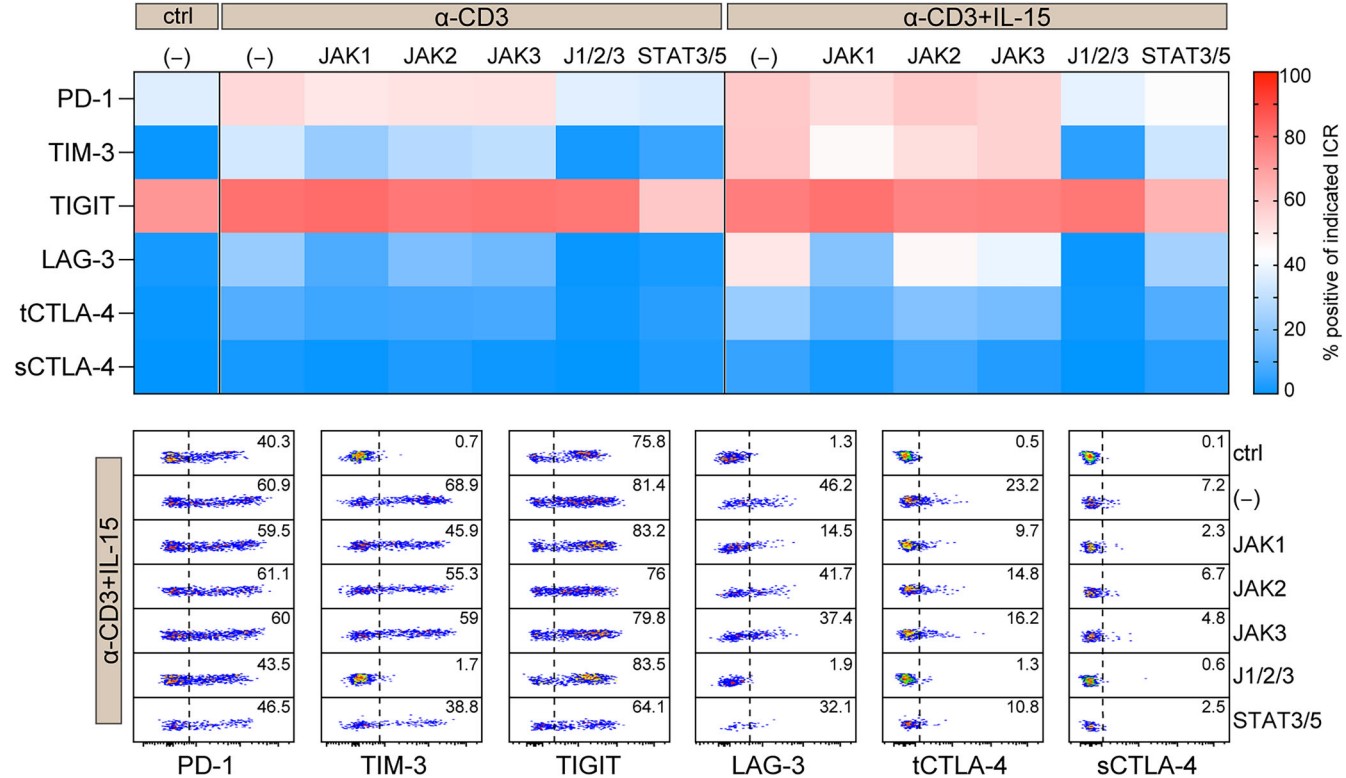

**B** Vδ2 cells

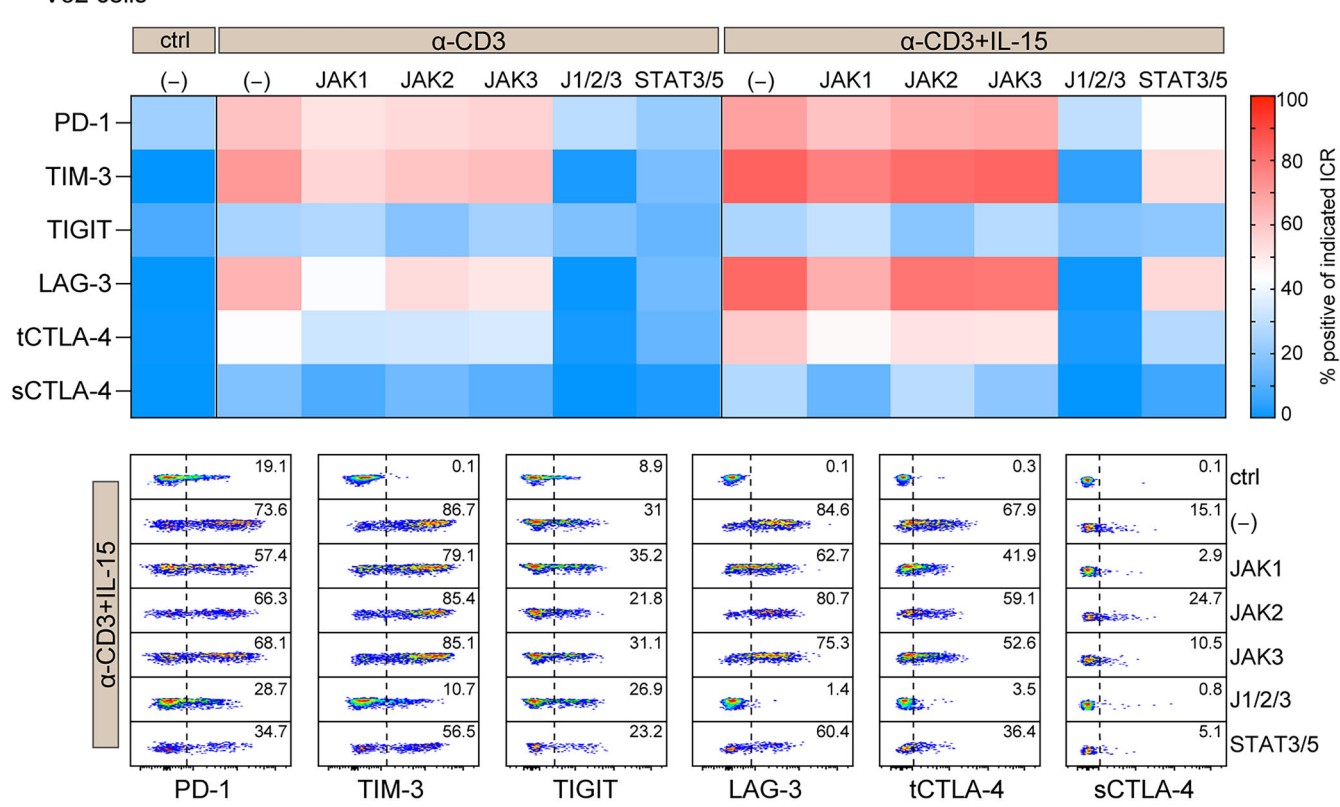

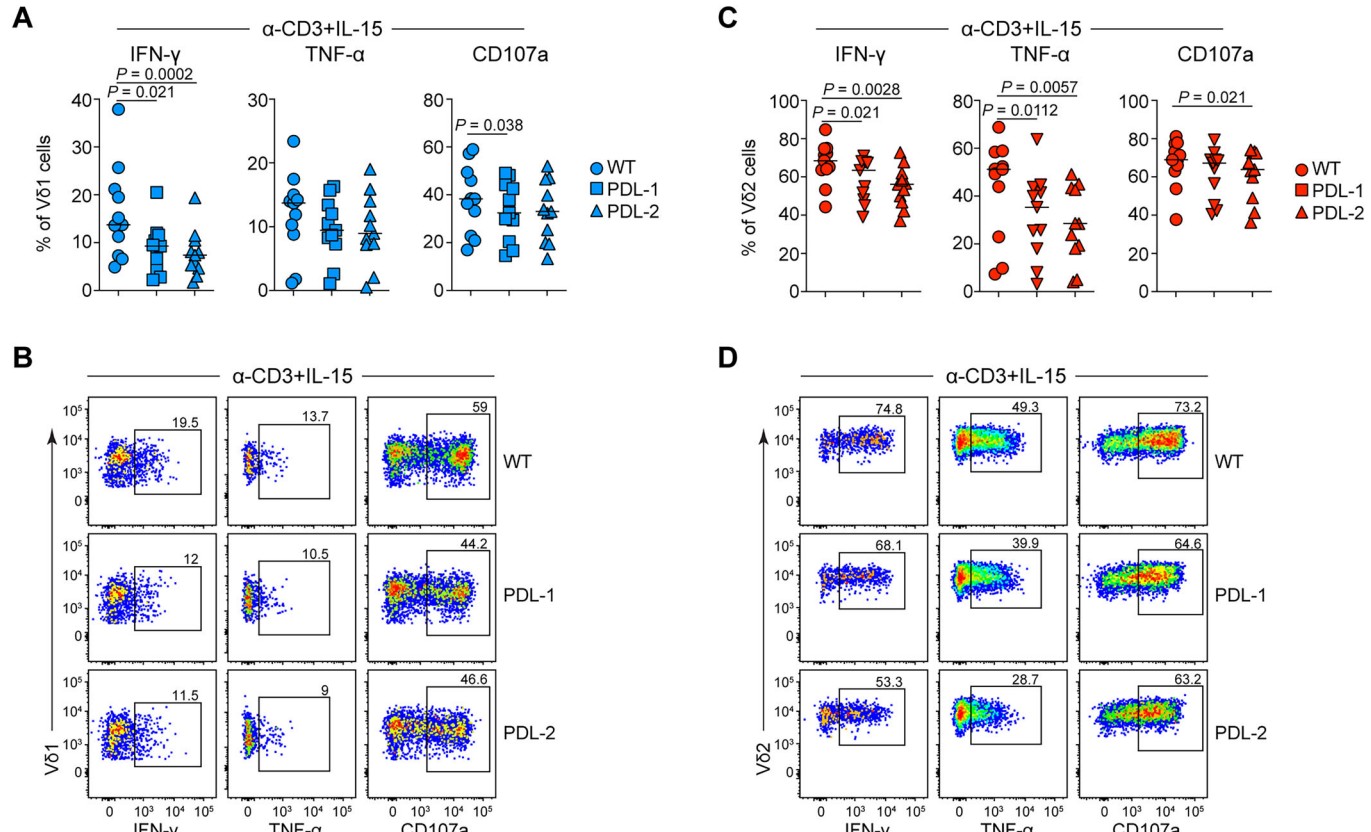

**Figure EV2.  PD-1 can inhibit the function of Vδ1 and Vδ2 cells.**

PBMCs were magnetically depleted of CD4+, CD19+ and CD14+ cells and cultured for 48 h with K562 cells expressing PDL-1, PDL-2 or no ligand (WT; wild-type), in the presence of IL-15 and α-CD3 stimulation. Levels of IFN-γ, TNF-α and CD107a were assessed by flow cytometry in Vδ1 (**A, B**) and Vδ2 (**C, D**) cells. Quantification (**A, C**) and representative plots (**B, D**) from 11 healthy donors in four independent experiments. *P* values were calculated by paired Friedman test with Dunn's multiple comparisons. In graphs, each symbol represents a donor.

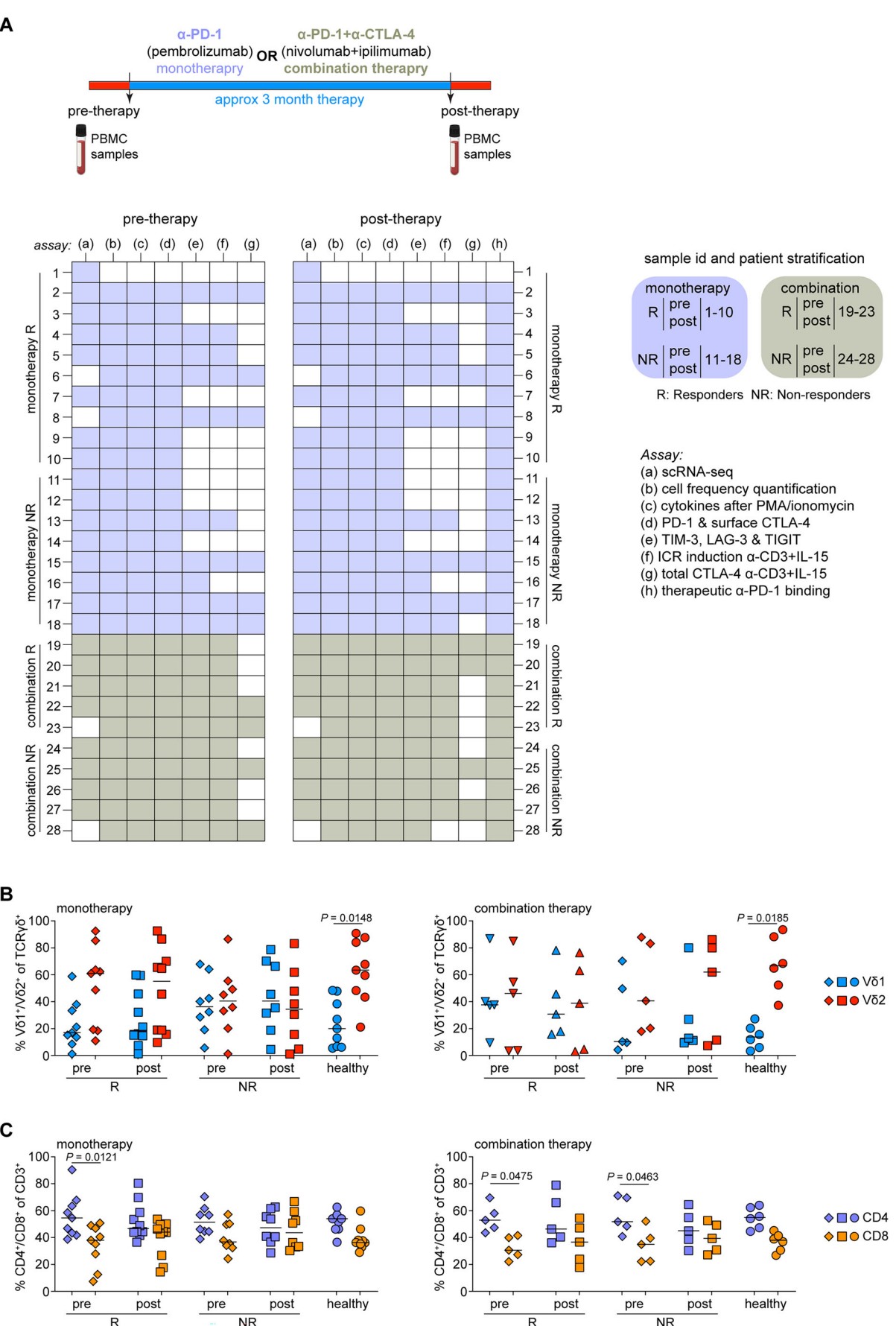

**Figure EV3.   ICB therapy study design and T cell frequencies in the various patient groups.**

(**A**) PBMC samples from patients with stage IV melanoma who were treated with either α-PD1 alone (pembrolizumab; monotherapy) or α-PD-1 and α-CTLA-4 (nivolumab +ipilimumab; combination therapy) were analyzed as shown. Samples were taken pre- and post-therapy from patients who responded (R) or did not respond (NR) to treatment. (**B**, **C**) Flow cytometry analysis depicting frequencies of Vδ1 (blue) and Vδ2 (red) cells (**B**) or CD4 (purple) and CD8 (orange) T cells (**C**) derived from patients after monotherapy (left) or after combination therapy (right). *P* values were calculated by paired two-way ANOVA and Tukey's multiple comparisons test. In graphs, each symbol represents a donor.

   

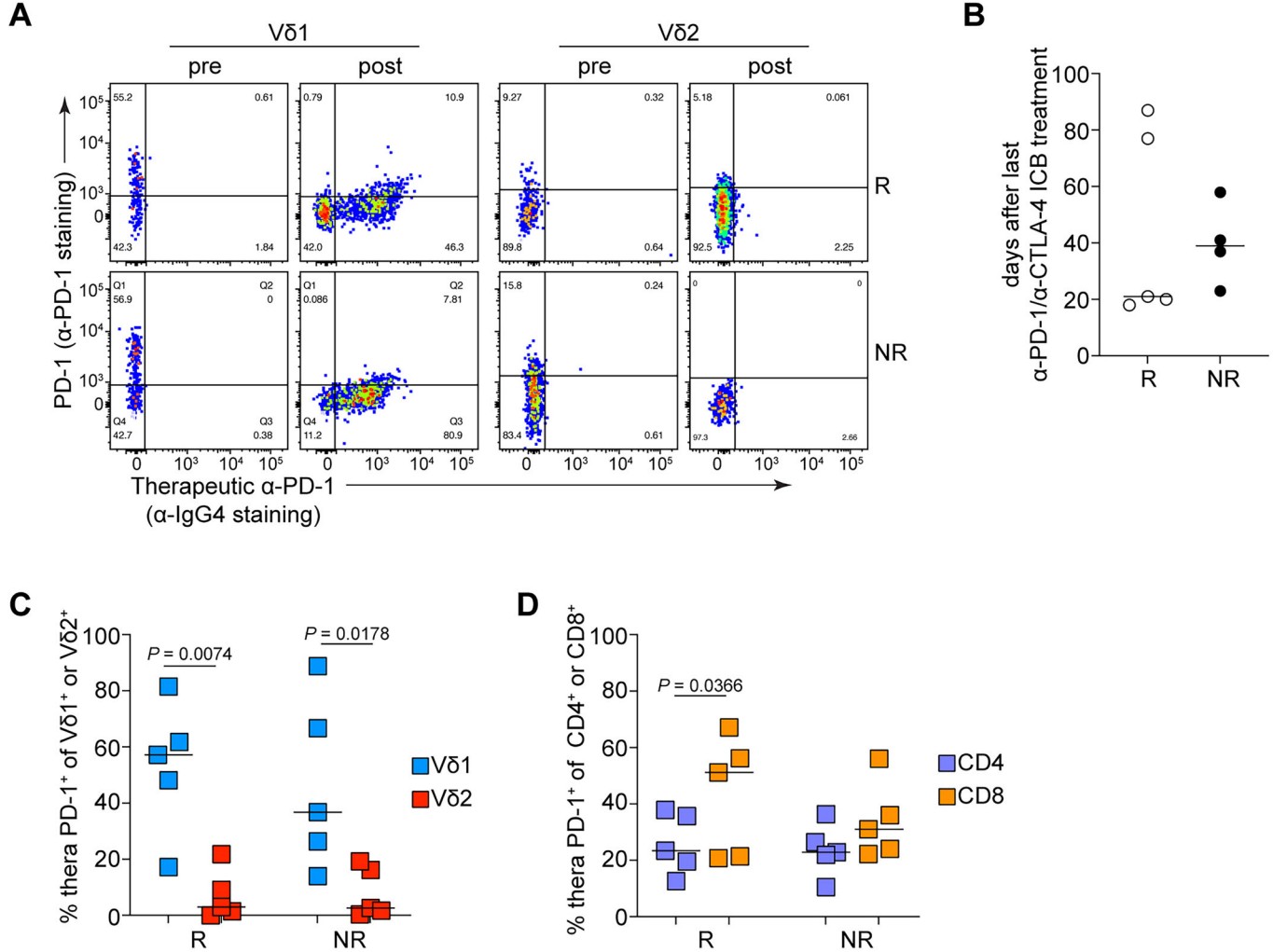

**Figure EV4. Therapeutic antibody binding in patients undergoing α-PD1 and α-CTLA-4 combination therapy.**

Analysis of PD-1 expression in PBMCs from patients with stage IV metastatic melanoma who either responded (R) or did not respond (NR) to a combination of nivolumab (therapeutic α-PD-1) and ipilimumab treatment. Paired samples were obtained from these patients before (pre) and 3 to 4 months after (post) the start of immunotherapy. The frequency of therapeutic antibody binding$^+$ (thera PD-1$^+$) cells was determined as explained in Fig. 2. (**A**) Representative flow cytometric analysis of therapeutic α-PD-1 antibody binding on the surface of Vδ1 and Vδ2 cells. (**B**) Days between the last antibody infusion and the collection of samples. (**C**) Thera PD-1$^+$ Vδ1 and Vδ2 cells after nivolumab+ipilimumab treatment. (**D**) Thera PD-1$^+$ CD4 and CD8 T cells after nivolumab+ipilimumab treatment. *P* values were calculated by paired two-way ANOVA and Tukey's multiple comparisons test. In graphs, each symbol represents a donor.

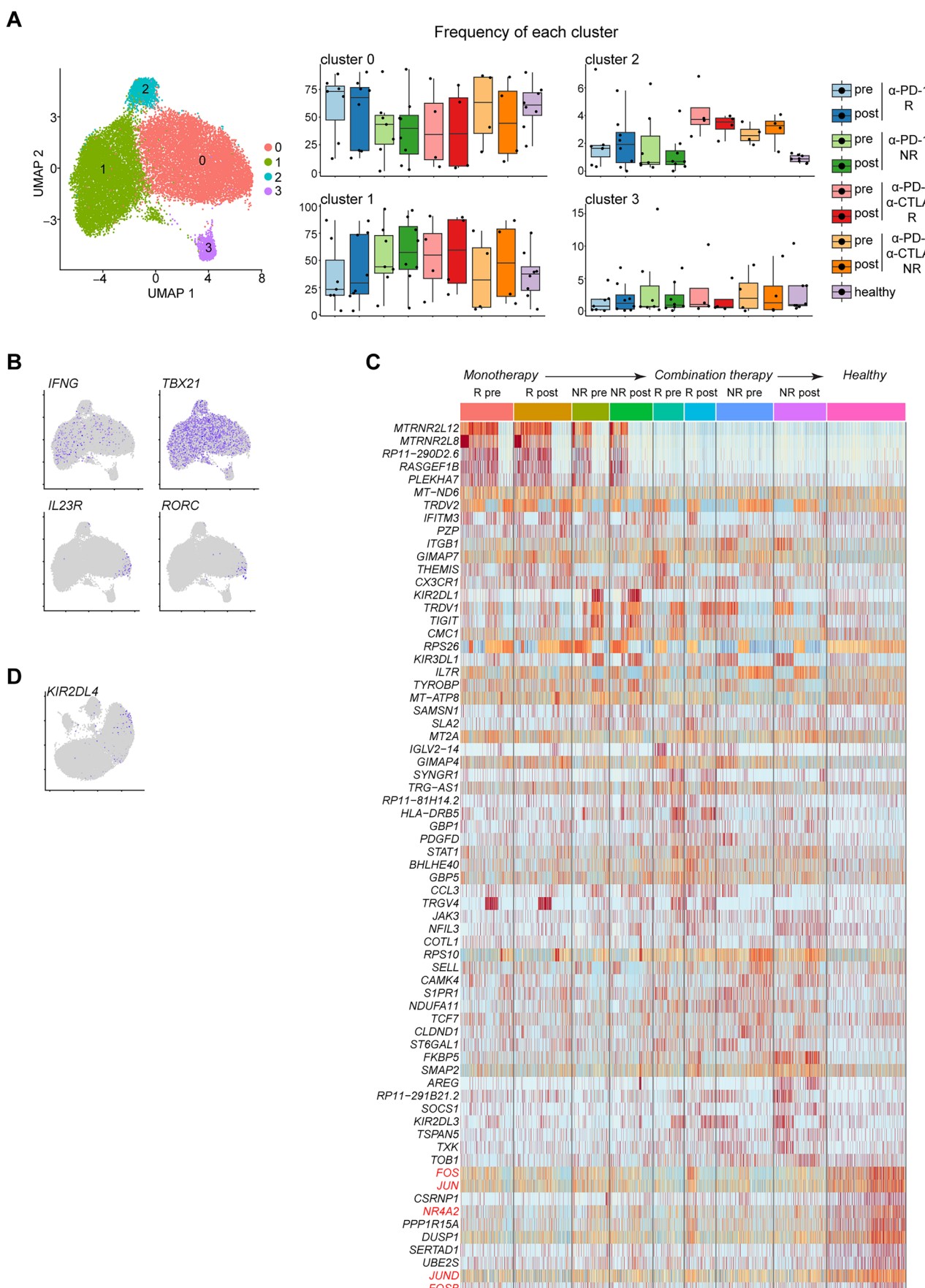

◀ **Figure EV5.   Transcriptomic analysis of circulating γδ T cells from patients with melanoma and healthy donors.**

(A) UMAP embedding showing clustering of circulating γδ T cells ($n = 84,537$) derived from healthy donors ($n = 8$) and patients with stage IV melanoma who received α-PD-1 ($n = 16$) or a combination of α-PD-1 and α-CTLA-4 ($n = 8$) treatment (left), and the distribution of cells between clusters (right). To calculate the frequency of cells assigned to each cluster we only considered samples with at least 50 cells; each dot represents an individual sample. Each box represents the interquartile range showing 25th percentile, median and 75th percentile, and whiskers extend to the minimum and maximum values in the dataset. (B) Distribution of *IFNG, TBX21, IL23R*, and *RORC* expression in the dataset. (C) Heatmap showing normalized expression of the 10 most differentially expressed genes (DEGs) between patient groups. R, responder; NR, non-responder; pre, samples were taken before the start of treatment; post, samples were taken 3 to 4 months after the start of treatment. DEGs were identified by Wilcoxon Rank Sum test as those with an average Log2 fold change of at least 0.5 and an adjusted $P < 0.05$. (D) Distribution of *KIR2DL4* expression in the dataset.

