## [Peer Review File · EMBO Molecular Medicine]

Dynamics of checkpoint receptors in $\gamma\delta$ T cell subsets are associated with clinical response during anti-PD-1 immunotherapies

Elisa Catafal-Tardós, Lola Dachicourt, Maria Virginia Baglioni, Marcelo Fares da Silva, Davide Secci, Marco Donia, Anders Kverneland, Inge Marie Svane, and Vasileios Bekiaris

Corresponding authors: Vasileios Bekiaris (vasbek@dtu.dk)

Review Timeline:

Submission Date:	26th Jun 25
Editorial Decision:	7th Jul 25
Appeal:	22nd Jul 25
Editorial Decision:	4th Sep 25
Revision Received:	19th Sep 25
Editorial Decision:	16th Oct 25
Revision Received:	29th Oct 25
Accepted:	30th Oct 25

Editor: Lise Roth

Transaction Report:

7th Jul 2025

Decision on your manuscript EMM-2025-22179

Dear Dr. Bekiaris,

Thank you for the submission of your manuscript to our journal following peer-review at a different venue. As agreed, I asked a single expert to review your revised manuscript, together with the point-by-point rebuttal letter.

I am sorry to report that, while he/she found the study interesting in principle, and agreed with your responses to the reviewers' criticisms, he/she also raised concerns about the small size of the cohort, the absence of an external validation cohort and the lack of rigor in the statistical tests performed.

Therefore, and after further internal discussion within the team, I am afraid that I have little choice but to return the manuscript to you at this time with the decision that we cannot offer to publish it.

I am very sorry to disappoint you in this occasion, and hope that the referees' comments are helpful in your continued work in this area.

Sincerely,

Lise Roth

Lise Roth
Senior Editor
EMBO Molecular Medicine

***** Reviewer's comments *****

Referee #1 (Comments on Novelty/Model System for Author):

Although I agree with the authors' response to the reviewers' criticisms, I have concerns about this manuscript due to the small size of the cohort ($n = 28$), which is divided into two treatment groups (anti-PD-1 ($n = 18$) and anti-PD-1 and anti-CTLA-4 ($n = 10$)) without external validation cohort. Methodologically, there are also various biases in the interpretation of the results, such as statistical analysis not taking into account multiple tests performed and confounding factors.

Referee #1 (Remarks for Author):

Compared to alpha beta T cells, the role of gamma-delta T cells in immunotherapy has received much less attention. This manuscript provides a detailed phenotypic and functional analysis of gamma-delta T cells before and after treatment with anti-PD-1 monotherapy or in combination with anti-CTLA-4.

Single-cell transcriptome analyses were conducted with the following controls: reasonable exclusion thresholds (less than 500 or more than 4000 genes, and less than 7% mitochondrial genes), elimination of doublets, and data integration using Harmony. However, the scope of the study in terms of identifying biomarkers of response to immunotherapy is limited by the small size of the analysed cohort ($n = 28$), which is divided into 18 patients receiving anti-PD-1 monotherapy and 10 receiving anti-PD-1 and anti-CTLA-4. In addition, there are no external validation cohorts.

Furthermore, the consideration of multiple analyses in the statistical analysis is not detailed. For scRNA-seq, the authors refer to 'adjusted $P < 0.05$ ' without specifying the method. For the other results (cytometry, transcriptomic analysis and Cox models), a Benjamini-Hochberg FDR correction should be performed if it has not already been done.

Some scRNA-seq analyses focus on very small subclusters ($\leq 2\%$ of $V\delta 1$).

Finally, the cohort is heterogeneous with respect to age, sex, cancer stage, the presence of BRAF mutations and PD-L1 status. It does not appear that these confounding factors have been taken into account. Additionally, there is a significant age difference between the anti-PD-1 monotherapy and combination therapy cohorts (72 vs. 55 years), which could introduce bias when interpreting the results.

Other comments

How should the binding of anti-PD-1 therapeutic antibodies to Vd1 be interpreted? Does this depend on the level of total PD-1 expression on these cells, or on the concentrations of anti-PD-1 in the blood?

Experiments show that, despite exhibiting a transcriptional exhaustion profile, $\gamma\delta$ T cells remain functional following polyclonal stimulation. It should be noted that PD-1, Tim-3 and Lag3 are also activation markers.

The lack of in situ analysis of these $\gamma\delta$ T lymphocyte populations also limits interpretation of their in vivo role in the immunotherapy mechanism of action. Are melanomas more infiltrated by gamma-delta 1 or gamma-delta 2 T cells in order to link the results to the tumour microenvironment of these patients?

The discussion could mention other markers that can predict resistance to anti-PD-1 independently of the combination with anti-CTLA-4 (Lucas et al., J Clin Oncol, 2023; Sam et al., EMBO Mol Med, 2025).

- The abbreviation 'Thera' should be defined.

As a service to authors, EMBO provides authors with the possibility to transfer a manuscript that one journal cannot offer to publish to another EMBO publication. The full manuscript and if applicable, reviewers reports are automatically sent to the receiving journal to allow for fast handling and a prompt decision on your manuscript. For more details of this service, and to transfer your manuscript to another EMBO title please click on Link Not Available

July 22nd, 2025

Dear Dr Lise Roth,

Thank you for considering our manuscript "*Checkpoint expression and therapeutic anti-PD-1 binding in V δ 1 T cells can discern effective cancer immunotherapy*" as an article in EMBO Molecular Medicine. As previously discussed, we would like to address the major criticisms by Referee #1 as communicated on 07/07/2025 and on 11/07/2025. The reviewer's comments are marked in **blue**, our responses are in **bold**.

Comments on 07/07/2025:

However, the scope of the study in terms of identifying biomarkers of response to immunotherapy is limited by the small size of the analysed cohort (n = 28), which is divided into 18 patients receiving anti-PD-1 monotherapy and 10 receiving anti-PD-1 and anti-CTLA-4. In addition, there are no external validation cohorts.

- We don't think the reviewer understood the scope of our study, which is not identifying biomarkers for immunotherapy. The scope of the study is to better understand the potential implications of $\gamma\delta$ T cells in the context of ICB and all the data that we show are novel and new for the community and the field. Thus, the cohort size in this regard is of little significance. There are many high-profile publications with much smaller cohort sizes. For example, a recent Nat Comm paper by the Ohashi lab was based on almost entirely one patient (<https://pubmed.ncbi.nlm.nih.gov/38321065/>). The study by Fong, Farber and colleagues in Nat Immunol in 2023 was based on relatively few samples (<https://pmc.ncbi.nlm.nih.gov/articles/PMC10063448/#Sec2>). Our discovery that V δ 1 cells bind high levels of therapeutic anti-PD-1 in patients who respond was completely serendipitous, which is why is at the end of our paper. We do indeed suggest that this biological property of V δ 1 cells could be used as a prognostic marker in the future but the study was not set up for such discoveries. This is a basic science paper that bridges clinical research and the data led us to a serendipitous finding.

Furthermore, the consideration of multiple analyses in the statistical analysis is not detailed. For scRNA-seq, the authors refer to 'adjusted P < 0.05' without specifying the method. For the other results (cytometry, transcriptomic analysis and Cox models), a Benjamini-Hochberg FDR correction should be performed if it has not already been done.

- We believe that we have gone into great details choosing and performing the most appropriate statistical tests for all analyses. All our tests are mentioned in the figure legends. We believe that this reviewer did not read the manuscript in its entirety.

Some scRNA-seq analyses focus on very small subclusters (**{less than or equal to}2% of V δ 1**). Finally, the cohort is heterogeneous with respect to age, sex, cancer stage, the presence of BRAF mutations and PD-L1 status. It does not appear that these confounding factors have been taken into account. Additionally, there is a significant age difference between the anti-PD-1 monotherapy and combination therapy cohorts (72 vs. 55 years), which could introduce bias when interpreting the results.

- We believe that the reviewer is referring to cluster c3 in Figure 5. This cluster expresses the highest levels of transcripts for ICRs, KIRs and TOX and it is indeed of low frequency. However, this matter has been discussed extensively and was not by any means overinterpreted. On the contrary, the potential importance of the cells comprising this cluster is substantially toned down. In fact, as we show in Fig 5F, in some patients 20-60% of all $\gamma\delta$ T cells are within this cluster, something that we think the reviewer also overlooked.

- We think that the comments regarding the heterogeneity of the cohort is overreaching and largely irrelevant in our study. Again, this is not a biomarker study and we believe that heterogeneity makes our mechanistic data stronger. Confounding factors in our analyses will make no sense as in many other studies like ours.

Comments on 11/07/2025:

Setting aside the biomarker aspect, the study remains descriptive of the regulation of ICI on gamma delta T cells. However, I don't see a clear main message or a potential clinical impact of these results.

- To the best of our knowledge this is the first mechanistic and systematic study of how ICRs are regulated in human peripheral $\gamma\delta$ T cells. We show: 1) distinct patterns of expression, identifying constitutive and inducible ICRs, 2) differential expression and induction in $\gamma\delta$ T cell subsets, 3) define cytokines that induce ICRs either alone or in combination with the TCR, 4) show which molecules of the JAK-STAT pathway are mostly involved in ICR induction following IL-15+TCR stimulation.

- Regarding the comment on the lack of potential clinical impact. The study has been and is supported and endorsed by the leadership of Danish National Center for Cancer Immunotherapy (CCIT) with enthusiasm due to its potential clinical application. Furthermore, an independent review panel by the Novo Nordisk Foundation funded this work partly because of its potential clinical application. As a testament of this, the next two pages contain personal letters by Professors Inge Marie Svane (head of CCIT) and Marco Donia, both physicians and global leaders in the field of cancer immunology and immunotherapy, who testify regarding the clinical importance of this study.

We are happy to provide more comments if necessary and are looking forward to hearing from you.

Yours faithfully,

Vasileios Bekiaris, on behalf of all listed co-authors

July 18 2025

EMBO Molecular Medicine

Dear Editor Lisa Roth,

I am reaching to share my clinical perspective on the study entitled "*Checkpoint expression and therapeutic anti-PD-1 binding in V δ 1 T cells can discern effective cancer immunotherapy*" which is based on work performed by Vasileios Bekiaris and his team involving clinical material from melanoma patients treated in my clinic.

The reason why I found it important to enter into this collaboration is because we are still lacking a full insight into factors affecting the efficacy of the checkpoint inhibitor based immune therapies being used for treatment of thousands of cancer patients every day. To further optimize clinical decision-making, we need to understand the full picture and especially, it is important to unravel parameters of impact which can potentially be clinical implemented as early indicators of response. This is indeed what appears to be the case in this study.

Further, this study has revealed a potential mechanistic role of gd- T cells in relation to checkpoint inhibitor therapy. This is indeed a new discovery with important clinical implications to be further explored not only for therapeutic strategies but also for the potential involvement in immune-related toxicity associated with checkpoint inhibitor therapy and development of future management strategies.

In summary, as a clinician I find that the results presented in this study have huge clinical implications which calls for further exploration.

Sincerely

Inge Marie Svane

EMBO Molecular Medicine

Herlev, July 15th 2025

Dear Dr. Roth,

I would like to share my clinical perspective on the study entitled “*Checkpoint expression and therapeutic anti-PD-1 binding in V δ 1 T cells can discern effective cancer immunotherapy.*”

The work provides novel insight into the immune correlates of response to checkpoint blockade therapy, identifying binding of therapeutic PD-1 antibodies to circulating V δ 1 $\gamma\delta$ T cells in patients with melanoma who respond to checkpoint immunotherapy. From a clinical standpoint, this introduces a measurable, blood-based parameter that could potentially serve as an early indicator of therapeutic benefit, something that is currently lacking in our practice.

Furthermore, the study reveals a complex and subset-specific pattern of immune checkpoint receptor regulation within $\gamma\delta$ T cells, especially the V δ 1 population, suggesting unexplored mechanisms of immune modulation relevant to therapy response and resistance.

These findings provide a strong foundation for future clinical research into $\gamma\delta$ T cell-directed biomarkers and therapeutic strategies.

Sincerely Yours

Marco Donia

4th Sep 2025

Dear Dr. Bekiaris, dear Vasileios,

Thank you for your e-mail asking us to reconsider our decision on your manuscript and for providing a point-by-point letter to the previous advisor's comments. Please accept my apologies for the delay in getting back to you, which is due to the annual leave of both referees and editorial staff.

As communicated and agreed previously, I have reached out to two new referees, who evaluated your manuscript afresh, but also had access to your rebuttal letter. As you will see below, they are supportive of your study, and we would thus like to invite you to submit your revised work to EMBO Molecular Medicine.

We are expecting your revised manuscript within three months, if you anticipate any delay, please contact us.

We require:

- 1) A .docx formatted version of the manuscript text (including legends for main figures, EV figures and tables). Please make sure that the changes are highlighted to be clearly visible.
- 2) Individual production quality figure files as .eps, .tif, .jpg (one file per figure). For guidance, download the 'Figure Guide PDF' (<https://www.embopress.org/page/journal/17574684/authorguide#figureformat>).
- 3) At EMBO Press we ask authors to provide source data for the main manuscript figures. You will receive a separate email with instructions for providing source data with your revised manuscript, including how to upload and organize the files.

Additional information on source data and instruction on how to label the files are available

- 4) A .docx formatted letter INCLUDING the reviewers' reports and your detailed point-by-point responses to their comments. As part of the EMBO Press transparent editorial process, the point-by-point response is part of the Review Process File (RPF), which will be published alongside your paper.
- 5) A complete author checklist, which you can download from our author guidelines (<https://www.embopress.org/page/journal/17574684/authorguide#submissionofrevisions>). Please insert information in the checklist that is also reflected in the manuscript. The completed author checklist will also be part of the RPF.
- 6) All Materials and Methods need to be described in the main text using our 'Structured Methods' format. According to this format, the Methods section includes a Reagents and Tools Table (listing key reagents, experimental models, software and relevant equipment and including their sources and relevant identifiers) followed by a Methods and Protocols section describing the methods, ideally using a step-by-step protocol format. The aim is to facilitate adoption of the methodologies across labs. Please download and fill our Reagents and Tools Table template (.docx), which you can find in our author guidelines: <https://www.embopress.org/page/journal/14693178/authorguide#structuredmethods>. When submitting your revised manuscript, please do not include the Reagents and Tools Table in the Methods section of the manuscript but upload it as a separate file choosing the file type "Reagent Table".
- 7) Please note that all corresponding authors are required to supply an ORCID ID for their name upon submission of a revised manuscript.
- 8) It is mandatory to include a 'Data Availability' section after the Materials and Methods. Before submitting your revision, primary datasets produced in this study need to be deposited in an appropriate public database, and the accession numbers and database listed under 'Data Availability'. Please remember to provide a reviewer password if the datasets are not yet public (see <https://www.embopress.org/page/journal/17574684/authorguide#dataavailability>).

9) For data quantification: please specify the name of the statistical test used to generate error bars and P values, the number (n) of independent experiments (specify technical or biological replicates) underlying each data point and the test used to calculate p-values in each figure legend. The figure legends should contain a basic description of n, P and the test applied. Graphs must include a description of the bars and the error bars (s.d., s.e.m.). Please provide exact p values.

10) Our journal encourages inclusion of *data citations in the reference list* to directly cite datasets that were re-used and obtained from public databases. Data citations in the article text are distinct from normal bibliographical citations and should directly link to the database records from which the data can be accessed. In the main text, data citations are formatted as follows: "Data ref: Smith et al, 2001" or "Data ref: NCBI Sequence Read Archive PRJNA342805, 2017". In the Reference list, data citations must be labeled with "[DATASET]". A data reference must provide the database name, accession number/identifiers and a resolvable link to the landing page from which the data can be accessed at the end of the reference. Further instructions are available at .

11) We replaced Supplementary Information with Expanded View (EV) Figures and Tables that are collapsible/expandable online. EV Figures should be cited as 'Figure EV1, Figure EV2' etc... in the text and their respective legends should be included in the main text after the legends of regular figures.

12) The paper explained: EMBO Molecular Medicine articles are accompanied by a summary of the articles to emphasize the major findings in the paper and their medical implications for the non-specialist reader. Please provide a draft summary of your article highlighting

13) Author contributions: CRedit has replaced the traditional author contributions section because it offers a systematic machine readable author contributions format that allows for more effective research assessment. Please remove the Authors Contributions from the manuscript and use the free text boxes beneath each contributing author's name in our system to add specific details on the author's contribution. More information is available in our guide to authors.

Please also suggest a visual abstract to illustrate your article as a PNG file 550 px wide x 300-600 px high. A cropped portion of this image will serve as thumbnail for the table of content on our webpage.

16) As part of the EMBO Publications transparent editorial process initiative (see our Editorial at <http://embomolmed.embopress.org/content/2/9/329>), EMBO Molecular Medicine will publish online a Review Process File (RPF) to accompany accepted manuscripts.

In the event of acceptance, this file will be published in conjunction with your paper and will include the anonymous referee reports, your point-by-point response and all pertinent correspondence relating to the manuscript. Let us know whether you agree with the publication of the RPF and as here, if you want to remove or not any figures from it prior to publication. Please note that the Authors checklist will be published at the end of the RPF.

I look forward to receiving your revised manuscript.

Yours sincerely,

Lise Roth

Please use this link to login to the manuscript system and submit your revision:
<https://embomolmed.msubmit.net/cgi-bin/main.plex>

***** Reviewer's comments *****

Referee #2 (Comments on Novelty/Model System for Author):

The manuscript by Catafal-Tardos et al. elucidates a area of tumor immunology, that has so far lain rather in darkness. Namely the role of gamma/delta T cells with respect to immune checkpoint receptors (ICR). The manuscript comprehensively describes the expression patterns and regulation of various ICR on different types of g/d T cells in healthy donors and melanoma patients under ICR-inhibitor (ICI) treatment. Substantial single cell sequencing corroborates the findings and clinical relevance is indicated by the observation, that under anti-PD1 monotherapy g/d only responders show binding of the therapeutic AB by g/d T cells.

I agree with the authors that this work is not a fishing expedition that scans large data sets for molecular signatures that can act a predictive or prognostic markers. Thus dataset size is not critical as is the need for a validation cohort.

I think, that this manuscript is of high relevance for the scientific community, as it provides various information about the behavior of g/D T cells and the results are valid and solid. The manuscript is indeed descriptive in its nature, however ex vivo experiments are performed and the gain of knowledge is substantial. Thus, I am in favor of publishing this manuscript in EMBO molecular medicine.

I also think that the authors have responded sufficiently to the comments from previous reviewers.

Referee #2 (Remarks for Author):

Despite my overall positive opinion, I have some minor point of criticism which should be addressed:

Figure 1 has unlabeled y-axes.

Figure 1A: The heat map is redundant, depiction of a set of representative histograms like in the other panels would be more informative.

Line 179:

The heading is misleading: the author refer to PD-1-mediated inhibition of g/d T cell function, but the heading can be interpreted as inhibition of PD-1. Thus it is confusing, when the PD-1 antibodies come into play, because normally, such antibodies are antagonists, but here they are coated and activate the receptors.

Line 212: To say that the frequencies of the cells are not affected is not fully correct.

Figure 4 B and D:

In panel D, TIGIT and the KIRs appear totally absent in c2, but according to panel B, there is some expression. Please check Figure 5E

There appears to be substantial differences in the cluster distribution between the Monotherapy group and the combination therapy group before treatment. C1 seems to be almost absent in the monotherapy group while both healthy and combi-therapy patient have it, before they are treated. Hence the intrinsic differences between both groups should be explained better.

Referee #3 (Comments on Novelty/Model System for Author):

The data is extensive, well-presented, relevant and novel. The authors have used a combination of standard techniques such as flow cytometry, ex vivo activation assays, and transcriptomic techniques to assess gamma-delta T cells in melanoma patients undergoing immunotherapy.

Referee #3 (Remarks for Author):

This paper extensively characterizes gamma-delta T cell subsets in human (healthy) and melanoma patients (before-after immunotherapy). The study focuses initially on the differential expression of immune checkpoint receptors, then proceeds to an extensive transcriptomic characterization. The data is clinically highly relevant. The techniques, experiments and conclusions are well presented and very useful for the biomedical researcher in cancer.

It is my opinion that it has sufficient clinical significance to be published in EMBO molecular medicine, pending a major reorganization. But I do not think that more experiments are needed. It provides a large amount of significant data for clinicians and researchers.

The weakness in this manuscript is the way that it is organized which hampers providing a clear point to the paper. To deal with this issue, I propose some changes to highlight its relevance.

1. The title: "Checkpoint expression and therapeutic anti-PD-1 binding in Vdelta1 T cells...". I do not think it truly reflects the main findings in the study, which heavily rely on the transcriptomic data. The main point of the study is something like "Dynamics of human gamma delta T cell subsets during anti-PD-1-based immunotherapies and clinical response to melanoma"

2. The most important data is shown in the transcriptomic profiles from all the subsets, some of it shown in figures from S6 to S8. I would suggest the authors to re-think the main figures.

For example, the first 1-to-3 figures deal with a standard phenotyping of cells and some in vitro functional assays. The data is certainly interesting, but it could well be re-organized to include some of it in supplementary figure, while the key findings be kept in a single figure, for example. Nevertheless, some of the ICR expression is also shown in the transcriptomics data.

3. The jak-stat data, while interesting, it is in my opinion secondary to the main findings in the paper. Yes, this pathway regulates some of the ICR expression. And? It ends here. It is interesting, but nothing would really change if it appears in supplementary. It would be much relevant if some specific cytokine profiles would appear in the patient either systemically or within the tumor microenvironment. Otherwise, in my opinion, it is just some addition here that deviates the attention from the main conclusions of the paper.

4. Another issue is the final part of the paper, the detection of therapeutic anti-PD-1 on gamma delta T cells. It looks out of place in the overall organization of the paper. It looks like another issue and another paper as it stands there, at the end of the manuscript. For example, do gamma-delta Vd1 cells in non-responders bind to anti-PD-1 and then endocytose it and degrade it? Or is there any other cell type/mechanism that depletes anti-PD-1 antibody? What is the concentration of the free anti-PD-1 antibody in responders vs non-responders? What about standard T cells? Surely we cannot present the data just only on gamma-delta T cells, when standard T cells are also relevant for response. As this is not a biomarker study, one would surely want to delve into the specifics of the mechanism.

If the authors are keen to show this data in this paper, they need to place it elsewhere or save it for another major paper. My sincere suggestion is to keep this very interesting data in the paper (it potentially has major implications), but to present this data much earlier to justify the rest of the paper. For example, something like this rationale: "We decided to look at gamma delta T cells in melanoma patients undergoing anti-PD1 therapy and combination with anti-CTLA4. Hence, we first wanted to demonstrate that anti-PD-1 can indeed bind gamma delta T cells in vivo in melanoma patients. To do that....Interestingly, we found this association between responders and PD-1 binding..... Then, we decided to explore the characteristics of gamma-delta T cells in patients through transcriptomic characterisation and the changes between cells from responder treated versus non-responder treated patients.....etc". It may provide some mechanistic differences in gamma-delta T cells leading to response.

Dear Dr Roth,

Below is our point-by-point response to the reviewers' comments. All our comments are in **bold**. All changes in the manuscript are marked in **yellow**. Please note that due to comments from Referee #3 we have re-structured the manuscript and most figure numbers have changed. The new figure numbers are indicated in each of our comments.

Reviewers' comments, following external review

Referee #2 (Comments on Novelty/Model System for Author):

The manuscript by Catafal-Tardos et al. elucidates a area of tumor immunology, that has so far lain rather in darkness. Namely the role of gamma/delta T cells with respect to immune checkpoint receptors (ICR). The manuscript comprehensively describes the expression patterns and regulation of various ICR on different types of g/d T cells in healthy donors and melanoma patients under ICR-inhibitor (ICI) treatment. Substantial single cell sequencing corroborates the findings and clinical relevance is indicated by the observation, that under anti-PD1 monotherapy g/d only responders show binding of the therapeutic AB by g/d T cells.

I agree with the authors that this work is not a fishing expedition that scans large data sets for molecular signatures that can act a predictive or prognostic markers. Thus dataset size is not critical as is the need for a validation cohort.

I think, that this manuscript is of high relevance for the scientific community, as it provides various information about the behavior of g/D T cells and the results are valid and solid. The manuscript is indeed descriptive in its nature, however ex vivo experiments are performed and the gain of knowledge is substantial. Thus, I am in favor of publishing this manuscript in EMBO molecular medicine.

I also think that the authors have responded sufficiently to the comments from previous reviewers.

Referee #2 (Remarks for Author):

Despite my overall positive opinion, I have some minor point of criticism which should be addressed:

Figure 1 has unlabeled y-axes.

- This has been now corrected.

Figure 1A: The heat map is redundant, depiction of a set of representative histograms like in the other panels would be more informative.

- We have replaced the heatmap with histograms.

Line 179:

The heading is misleading: the author refer to PD-1-mediated inhibition of g/d T cell function, but the heading can be interpreted as inhibition of PD-1. Thus it is confusing, when the PD-1 antibodies come into play, because normally, such antibodies are antagonists, but here they are coated and activate the receptors.

- The reviewer is correct. We have changed this heading as follows: In vitro PD-1-mediated inhibition assays have minimal impact on human $\gamma\delta$ T cell function. The change is highlighted in the text; see line 182.

Line 212: To say that the frequencies of the cells are not affected is not fully correct.

- We wrote this because we could not detect statistical differences among treatment groups. We have now incorporated in our text the observed differences among T cell subsets within each treatment group; see line 214.

Figure 4 B and D:

In panel D, TIGIT and the KIRs appear totally absent in c2, but according to panel B, there is some expression. Please check

- In panel B we show expression of these markers in the individual cells of this cluster. On the other hand, in panel D we show the global expression of these markers in the entire cluster. Since only a very small proportion of the cells in c2 express TIGIT and KIRs this does not show up in the violin plots in panel D. In our experience this is very common with these kind of data. Mind that Figure 4 is now Figure 3.

Figure 5E

There appears to be substantial differences in the cluster distribution between the Monotherapy group and the combination therapy group before treatment. C1 seems to be almost absent in the monotherapy group while both healthy and combi-therapy patient have it, before they are treated. Hence the intrinsic differences between both groups should be explained better.

- We agree with the reviewer and we have addressed this patient group heterogeneity when we describe Vδ1, Vδ2, and Vδ3 clusters in our text (lines 310-312) and new supplementary figure Appendix Fig. S6A-C.

Referee #3 (Comments on Novelty/Model System for Author):

The data is extensive, well-presented, relevant and novel. The authors have used a combination of standard techniques such as flow cytometry, ex vivo activation assays, and transcriptomic techniques to assess gamma-delta T cells in melanoma patients undergoing immunotherapy.

Referee #3 (Remarks for Author):

This paper extensively characterizes gamma-delta T cell subsets in human (healthy) and melanoma patients (before-after immunotherapy). The study focuses initially on the differential expression of immune checkpoint receptors, then proceeds to an extensive transcriptomic characterization. The data is clinically highly relevant. The techniques, experiments and conclusions are well presented and very useful for the biomedical researcher in cancer.

It is my opinion that it has sufficient clinical significance to be published in EMBO molecular medicine, pending a major reorganization. But I do not think that more experiments are needed. It provides a large amount of significant data for clinicians and researchers.

The weakness in this manuscript is the way that it is organized which hampers providing a clear point to the paper. To deal with this issue, I propose some changes to highlight its relevance.

1. The title: "Checkpoint expression and therapeutic anti-PD-1 binding in Vdelta1 T cells...". I do not think it truly reflects the main findings in the study, which heavily rely on the transcriptomic data. The main point of the study is something like "Dynamics of human gamma delta T cell subsets during anti-PD-1-based immunotherapies and clinical response to melanoma"

- We really like the suggestion from the reviewer and as such we tried to combine the above suggestion with the original title. Our new title is: "Dynamics of checkpoint receptors in $\gamma\delta$ T cell subsets are associated with clinical response during anti-PD-1 immunotherapies".

2. The most important data is shown in the transcriptomic profiles from all the subsets, some of it shown in figures from S6 to S8. I would suggest the authors to re-think the main figures.

For example, the first 1-to-3 figures deal with a standard phenotyping of cells and some in vitro functional assays. The data is certainly interesting, but it could well be re-organized to include some of it in supplementary figure, while the key findings be kept in a single figure, for example. Nevertheless, some of the ICR expression is also shown in the transcriptomics data.

- We thank the reviewer for this suggestion. We have now reorganized the data so the previous figures S7 and S8 are now main figures 5 and 6, while previous main figures 2 and 3 are now supplementary figures EV1 and EV2.

3. The jak-stat data, while interesting, it is in my opinion secondary to the main findings in the paper. Yes, this pathway regulates some of the ICR expression. And? It ends here. It is interesting, but nothing would really change if it appears in supplementary. It would be much relevant if some specific cytokine profiles would appear in the patient either systemically or within the tumor microenvironment. Otherwise, in my opinion, it is just some addition here that deviates the attention from the main conclusions of the paper.

- These data is now in supplementary figure EV1.

4. Another issue is the final part of the paper, the detection of therapeutic anti-PD-1 on gamma delta T cells. It looks out of place in the overall organization of the paper. It looks like another issue and another paper as it stands there, at the end of the manuscript. For example, do gamma-delta Vd1 cells in non-responders bind to anti-PD-1 and then endocytose it and degrade it? Or is there any other cell type/mechanism that depletes anti-PD-1 antibody? What is the concentration of the free anti-PD-1 antibody in responders vs non-responders? What about standard T cells? Surely we cannot present the data just only on gamma-delta T cells, when standard T cells are also relevant for response. As this is not a biomarker study, one would surely want to delve into the specifics of the mechanism.

If the authors are keen to show this data in this paper, they need to place it elsewhere or save it for another major paper.

My sincere suggestion is to keep this very interesting data in the paper (it potentially has major implications), but to present this data much earlier to justify the rest of the paper. For example, something like this rationale: "We decided to look at gamma delta T cells in melanoma patients undergoing anti-PD1 therapy and combination with anti-CTLA4. Hence, we first wanted to demonstrate that anti-PD-1 can indeed bind gamma delta T cells in vivo in melanoma patients. To do that....Interestingly, we found this association between responders and PD-1 binding..... Then, we decided to explore the characteristics of gamma-delta T cells in patients through transcriptomic characterisation and the changes between cells from responder treated versus non-responder treated patients.....etc". It may provide some mechanistic differences in gamma-delta T cells leading to response.

- We agree with the reviewer. We originally followed that line of thought and had the above suggested structure. We changed the structure after we were advised that the anti-PD-1 binding data were the main driving message of this paper. We are happy to see the original structure of the paper restored. As such, we have made the following changes:

1) The section "Persistent binding of therapeutic antibody on Vδ1 cells discerns patients with melanoma who respond to α-PD-1 monotherapy" begins our studies in melanoma (lines 202-203)

2) This is followed by the scRNA-seq data (lines 254-255)

3) This is followed by the functional data (lines 348-349).

Because of this change in structure, we have amalgamated the sections "γδ T cells from patients with melanoma are functional following polyclonal stimulation" and "Expression and induction of ICRs in γδ T cells from patients with melanoma" into "γδ T cells from patients with melanoma are functional following polyclonal stimulation but show impaired induction of LAG-3 and CTLA-4" (see lines 348-385). As a result, the numbering of most figures has now changed.

16th Oct 2025

Dear Dr. Bekiaris, Dear Vasileios,

Thank you for submitting your revised study. We have now received the reports from the referees and as you will see below, they are satisfied with the revisions. I will therefore be able to accept your manuscript once the following editorial concerns are addressed:

1/ Manuscript text:

- Please remove the yellow highlights, and only indicate in track changes mode any new modification in the text.
- Please correct the section headings and order to: Abstract / Keywords / The Paper Explained / Introduction / Results / Discussion / Methods / Data Availability / Acknowledgements / Disclosure and Competing Interests Statement / References / Figure Legends / Expanded View Figure Legends
- Please provide up to 5 keywords.
- Materials and Methods should be changed to Methods:
 - o Human research participants: If collected and within the bounds of privacy constraints, report on age, sex and gender or ethnicity for all study participants (appendix table S1?). Please include a statement confirming that informed consent was obtained from all subjects and that the experiments conformed to the principles set out in the WMA Declaration of Helsinki and the Department of Health and Human Services Belmont Report.
 - o Cells: please indicate the origin of the cell lines, and provide a statement on authentication and mycoplasma contamination.
 - o Statistics: please provide a statement on randomization and blinding.
- "Declaration of interests" should be changed to "Disclosure statement and competing interests".
- The reference format should be changed to alphabetical order, with 10 authors listed before et al.

2/ Figures:

- Please make sure that all figures and figure panels are referenced in the text, chronologically (currently, Fig 5A and 6A are called out before Fig 4B and Fig 5B and 6B before Fig 4C; there is a callout for a Fig. S2, please correct).
- Please note that we can accommodate more than 5EV figures, and you are thus welcome to make some of your Appendix figures EV figures.
- Appendix: please remove and rename Appendix Table S2 to "Dataset EV1", remove the legend from the appendix file and add it to the excel file, in a separate tab/worksheet.
- Please address the queries from our data editors in the figure legends:
 1. Please define the annotated p values ****/***/**/* as well as provide the exact p-values for the same in the legend of figure 2D as appropriate.
 2. Please note that the exact p values are not provided in the legends of figures 1A-G; 7A, B, E; 8A, B, C, E, F, G; EV2 A, C; EV3 B, C; EV4 C, D; EV5 A.
 3. Please indicate the statistical test used for data analysis in the legends of figures 2D, G, H; 7A, B, C, E, F, G.
 4. Please note that the box plots need to be defined in terms of minima, maxima, centre, bounds of box and whiskers, and percentile in the legend of figure EV5 A
 5. Please note that information related to n is missing in the legends of figures 2D, 3D, 4C, D; 7A, B, C, E, F, G

3/ Checklist:

- Cell materials: please fill in the subsection on authentication and mycoplasma contamination.
- Human research participants: If collected and within the bounds of privacy constraints report on age, sex and gender or ethnicity for all study participants (Appendix Table S1?).
- Experimental study design and statistics: please fill in the subsections on blinding and randomization.

4/ I introduced minor changes in your Paper Explained. Please let me know if you agree or amend as you see fit, and add the text to the main manuscript text file.

Problem

Immune checkpoint blockade (ICB) therapies targeting the PD-1 receptor are widely used to treat various cancers. Although ICB therapies indiscriminately target all cells that express the immune checkpoint receptor (ICR), most studies have focused on conventional CD4 and CD8 T cells. $\gamma\delta$ T cells are polyfunctional lymphocytes that can be exceptionally effective at killing cancers and are therefore potential targets of ICB. However, $\gamma\delta$ T cells have only recently begun to be systematically studied in relation to ICRs and ICB.

Results

We identified constitutive and inducible ICR expression patterns in human $\gamma\delta$ T cell subsets, then studied their response to anti-PD-1-based immunotherapy. We analyzed circulating $\gamma\delta$ T cells from melanoma patients and demonstrated that different subsets exhibited distinct transcriptional and functional characteristics associated with cancer and clinical responses to ICB. We found

that, in patients who successfully responded to anti-PD-1 therapy, V δ 1 $\gamma\delta$ T cells specifically bound high levels of therapeutic antibody.

Impact

The differential expression of ICRs by $\gamma\delta$ T cell subsets could provide insight into the types of responses that will be elicited by different ICB therapies. Furthermore, the specific biological properties of $\gamma\delta$ T cells, such as the persistent binding of the anti-PD-1 therapeutic, could be useful in predicting the efficacy of ICB therapy for individual patients.

5/ Synopsis:

I introduced minor modifications in your synopsis, please let us know if you agree or amend as you see fit:

"Different subsets of $\gamma\delta$ T cells display differential expression of immune checkpoint receptors (ICRs) and have distinct biological characteristics that are associated with cancer and the efficacy of immune checkpoint blockade therapy.

- In V δ 1 and V δ 2 T cells, PD-1 and TIGIT are constitutively expressed, whereas TIM-3, LAG-3 and CTLA-4 are induced upon activation.
- TIGIT and CTLA-4 show preferential high expression in V δ 1 or V δ 2 cells, respectively.
- In melanoma, both V δ 1 and V δ 2 cells show reduced expression of AP-1 genes and impaired LAG-3 and CTLA-4 induction, while V δ 1 cells express high levels of ICR and KIR genes.
- Successful anti-PD-1 monotherapy was associated with lower expression levels of KIR genes and persistent therapeutic anti-PD-1 binding on V δ 1 cells."

Thank you for providing a nice visual abstract, please resize it as a jpeg/tiff/PNG file 550 px wide x 300-600 px high. A cropped portion of this image will serve as thumbnail for the table of content on our webpage.

6/ As part of the EMBO Publications transparent editorial process initiative (see our Editorial at <http://embomolmed.embopress.org/content/2/9/329>), EMBO Molecular Medicine will publish online a Review Process File (RPF) to accompany accepted manuscripts.

This file will be published in conjunction with your paper and will include the anonymous referee reports, your point-by-point response and all pertinent correspondence relating to the manuscript. Let us know whether you agree with the publication of the RPF and as here, if you want to remove or not any figures from it prior to publication.

I look forward to receiving your revised manuscript.

Yours sincerely,

Lise Roth

To submit your manuscript, please follow this link:

<https://embomolmed.msubmit.net/cgi-bin/main.plex>

***** Reviewer's comments *****

Referee #2 (Comments on Novelty/Model System for Author):

The manuscript by Catafal-Tardos et al. elucidates a area of tumor immunology, that has so far lain rather in darkness. Namely the role of gamma/delta T cells with respect to immune checkpoint receptors (ICR). The manuscript comprehensively describes the expression patterns and regulation of various ICR on different types of g/d T cells in healthy donors and melanoma patients under ICR-inhibitor (ICI) treatment. Substantial single cell sequencing corroborates the findings and clinical relevance is indicated by the observation, that under anti-PD1 monotherapy g/d only responders show binding of the therapeutic AB by g/d T cells.

I agree with the authors that this work is not a fishing expedition that scans large data sets for molecular signatures that can act a predictive or prognostic markers. Thus dataset size is not critical as is the need for a validation cohort.

I think, that this manuscript is of high relevance for the scientific community, as it provides various information about the behavior of g/D T cells and the results are valid and solid. The manuscript is indeed descriptive in its nature, however ex vivo experiments

are performed and the gain of knowledge is substantial. Thus, I am in favor of publishing this manuscript in EMBO molecular medicine.

The authors have responded sufficiently to my previous comments and have improved the manuscript accordingly.

Referee #2 (Remarks for Author):

The authors have addressed my comments sufficiently and have corrected the manuscript accordingly. Hence I deem the manuscript suitable for publication

Referee #3 (Comments on Novelty/Model System for Author):

The manuscript is technically and scientifically top. I has also undergone many revision rounds, so that it has been extensively reviewed.

Referee #3 (Remarks for Author):

The authors have addressed all the points raised by the reviewer. Following the reorganization of the manuscript, is significantly more attractive and clear to EMBO's audience. This is a top manuscript that is suitable for publication after so many revisions.

The authors addressed the remaining editorial issues.

30th Oct 2025

Dear Dr. Bekiaris, Dear Vasileios,

Thank you for submitting your revised files. I am pleased to inform you that your manuscript is accepted for publication and is now being sent to our publisher to be included in the next available issue of EMBO Molecular Medicine.

With kind regards,

Lise
